# Comparison of CRISPR-Cas13b RNA base editing approaches for USH2A-associated inherited retinal degeneration
Lewis E. Fry [1,2,3], Lauren Major[1], Ahmed Salman [1], Lucy A. McDermott[1], Jun Yang [4], Andrew J. King [5], Michelle E. McClements [1] & Robert E. MacLaren [1,6] ✉

CRISPR-Cas13 systems have therapeutic promise for the precise correction of point mutations in RNA. Using adenosine deaminase acting on RNA (ADAR) effectors, A-I base conversions can be targeted using guide RNAs (gRNAs). We compare the Cas13 effectors PspCas13b and Cas13bt3 for the repair of the gene *USH2A*, a common cause of inherited retinal disease and Usher syndrome. In cultured cells, we demonstrate up to 80% efficiency for the repair of the common c.11864 G > A and its murine equivalent c.11840 G > A, across different gRNAs and promoters. We develop and characterize a mouse model of Usher syndrome carrying the c.11840 G > A mutation designed for the evaluation of base editors for inherited retinal disease. Finally, we compare Cas13 effectors delivered via AAV for the repair of *Ush2a* in photoreceptors. Mean RNA editing rates in photoreceptors across different constructs ranged from 0.32% to 2.04%, with greater efficiency in those injected with PspCas13b compared to Cas13bt3 constructs. In mice injected with PspCas13b constructs, usherin protein was successfully restored and correctly localized to the connecting cilium following RNA editing. These results support the development of transcriptome targeting gene editing therapies for retinal disease.

Programmable RNA base editing is an emerging therapeutic approach for the correction of point mutations in the transcriptome. Targeting RNA offers potential safety advantages to DNA base editing methods, as RNA edits are non-heritable, transient and reversible, and only occur in cells where the gene is expressed[1]. Methods of site-directed RNA editing broadly comprise two effectors: an effector to selectively bind RNA targets, and an effector to precisely change a single base[1].

CRISPR (Clustered Regularly Interspaced Short Palindromic Repeats) RNAs and CRISPR-associated (Cas13) proteins are an RNA-guided and RNA-targeting type VI CRISPR system. Cas13 proteins interact and bind RNA targets directed by a guide RNA (gRNA) and contain higher eukaryotes and prokaryotes nucleotide-binding (HEPN) domains, that exclusively cleave ssRNA[2]. Following the first identified Cas13 orthologue C2c2[3,4] (later renamed to Cas13a), a range of further orthologues have been identified[5]. Subsequently, nuclease-inactive "dead" Cas13 (dCas13) have been engineered through mutagenesis of the HEPN domains to abolish Cas13 RNase activity while preserving the ability of Cas13 to bind to RNA

directed by a gRNA. This enables dCas13 to be used as a programmable RNA binding molecule and to fuse effector proteins to dCas13 to undertake catalytic functions.

Adenosine deaminases acting on RNA (ADAR) are endogenously expressed proteins in human cells that mediate adenosine-inosine (A > I) post-transcriptional modifications in regions of double-stranded RNA[6]. The deaminase domain of the ADAR2 (ADAR2_{DD}) protein has since been engineered for use in a range of site-directed RNA editing systems[7]. As inosine is recognized as guanine in cellular processes, such as translation and splicing, A > I editing can, therefore, be used to correct G > A mutations, which make up 28% of pathogenic single-nucleotide variants reported on ClinVar[8]. Engineered ADAR2_{DD} mutants alter editing efficiency and specificity. The ADAR2_{DD}-E488Q mutation markedly increases editing activity[9–11], however it increases off-target editing in the transcriptome[12,13]. The ADAR2_{DD}-E488Q/T375G double mutant has demonstrated promise for greater specificity although with some loss of on-target efficiency[12].

[1]Nuffield Department of Clinical Neurosciences & NIHR Oxford Biomedical Research Centre, University of Oxford, Oxford, UK. [2]Royal Victorian Eye and Ear Hospital, East Melbourne, VIC, Australia. [3]Centre for Eye Research Australia, East Melbourne, VIC, Australia. [4]Department of Ophthalmology and Visual Sciences, John A. Moran Eye Center, University of Utah, Salt Lake City, UT, USA. [5]Department of Physiology, Anatomy and Genetics, University of Oxford, Oxford, UK. [6]Oxford Eye Hospital, Oxford University Hospitals NHS Foundation Trust, Oxford, UK. ✉e-mail: maclaren@eye.ox.ac.uk

Programmable RNA editing using CRISPR-Cas13 was first described in 2017 as the REPAIR (RNA Editing for Programmable Adenosine to Inosine Replacement) system, fusing nuclease-deactivated Cas13b ortholog from bacterium *Prevotella spp* (dPspCas13b) with ADAR2$_{DD}$[12]. Binding of the Cas13-ADAR-gRNA complex creates a region of double-stranded RNA at the gRNA binding site, allowing the ADAR2$_{DD}$ to mediate A-I base conversion[12]. The targeted A base is specified by an A-C mismatch encoded within the gRNA sequence. Successful editing of RNA transcripts in culture has since been reported by a number of groups[12,14–19]. The large size of PspCas13b restricts packaging with adeno-associated viral (AAV) vectors, and so more minimal Cas13-ADAR effectors with smaller coding sequences have been described[16,18,19], including a C-terminally truncated dPspCas13b (dPspCas13b-Δ984-1090)[12] and dCas13bt-ADAR[16].

Inherited retinal diseases (IRDs) are the most common cause of irreversible sight loss in people of working age, affecting approximately 1 in 2000 people[20,21]. Pathogenic variants in the *USH2A* gene are a leading cause of IRDs[22]. *USH2A* is associated with autosomal recessive non-syndromic retinitis pigmentosa, as well as Usher syndrome type II, characterized by congenital sensorineural hearing loss and retinitis pigmentosa[23]. The large (15.6 kb) coding sequence of the *USH2A* gene that encodes the usherin protein greatly exceeds the ~4.7 kb capacity of an AAV vector, and alternative treatment strategies for AAV gene replacement are required. Correction of common G > A mutations with RNA base editing is one such approach[24]. The c.11864 G > A (p. W3955X) mutation produces a premature termination codon (PTC) in exon 61 and is the third most common mutation in *USH2A*, accounting for 3–5% of cases worldwide[25–27]. It is particularly common in European populations where it has been implicated in 20–84% of affected families across different cohorts[28,29].

Whether RNA base editors can correct the transcriptome of mature photoreceptors in vivo is an outstanding question for which appropriate animal models are required. *Ush2a*$^{-/-}$ knockout mice with a deletion of exon 5 demonstrate a lack of retinal usherin expression and develop late-onset slowly progressive retinal degeneration and dysfunction[30]. The orthologues of *USH2A* (isoform B, 15,609 bp, 72 exons, 5202aa, ~590 kDA) and Ush2a (15,582 bp, 71 exons, 5192aa, ~590 kDA) expressed in the retina are highly conserved between species, with a 78% nucleotide identity, and a 71% amino acid identity. The homologous tryptophan residue implicated in the human p.W3955X mutation occurs at position 3947 in exon 60 of Ush2a. We hypothesized, therefore, that installation of a Ush2a c.11840 G > A (p.W3947X) mutation would generate a Ush2a-null phenotype and generate a useful model for the study of programmable base editing of Ush2a in vivo.

In this study, we developed and compared AAV-delivered dCas13b-ADAR systems capable of correcting G > A mutations a Ush2a. We first compared the dPspCas13b-ADAR[12] and the dCas13bt-ADAR[16] systems in a surrogate luciferase assay in cells in culture for the repair of mutations in both *USH2A* and *Ush2a*. We then developed a *Ush2a*$^{W3947X/W3947X}$ mouse for the study of base editing of *Ush2a* and compared AAV-delivered dCas13b-ADAR editing constructs in vivo for the correction of the *Ush2a* transcript and restoration of usherin expression in photoreceptors of the *Ush2a*$^{W3947X/W3947X}$ mouse.

## Results

### In vitro comparison *of* Cas13-ADAR orthologues for editing of *USH2A* and *Ush2a*

To investigate the potential for RNA editing systems to repair mutations in the RNA of retinal genes, we compared the efficiency of Cas13-ADAR orthologues for the correction of the human *USH2A* c.11864 G > A (p.W3955X) and the mouse *Ush2a* c.11840 G > A (p.W3947X) premature termination codons in cells in culture. As *USH2A* expression is confined to photoreceptors and inner hair cells and not expressed in commonly cultured cell lines, a dual luciferase assay containing a 200 bp target sequence of interest from *USH2A* and *Ush2a* was developed to report A-I editing activity for screening of constructs and gRNAs in HEK293T cells (Fig. 1A, B).

We firstly sought to characterize the dPspCas13b-ADAR$_{DD}$ system. gRNAs targeting *USH2A* p.W3955X and *Ush2a* p.W3947X were screened using the dPspCas13b(Δ984-1090)-ADAR$_{DD}$(E488Q), *n* (Fig. 1C). 50nt gRNAs were tiled across the target mutation with an A–C mismatch encoded at the target adenosine. The gRNAs varied by the distance between the mismatched base and the gRNA scaffold (mismatch distance, range 18–42 nts) (Fig. 1C). For the *USH2A* target, a guide with a 36nt mismatch distance (50 nt-36) was most active with a 38% editing rate (Fig. 1D). Screening the *Ush2a* target, editing rates were generally poor ( < 20%), except for a guide with a 24 nt mismatch distance (mUsh2a-24-G10) with an editing rate of 50% (Fig. 1E). The limited number of 30nt guides screened showed lower editing efficiency relative to the 50 nt guides (Fig. 1E).

We next sought to evaluate Cas13bt orthologues, as these are shorter and thus more amenable to packaging in AAV[16,19]. Three of these orthologues (Cas13bt1, Cas13bt3, and Cas13bt5) linked to ADAR$_{DD}$(E488Q) domains were tested with both 30 nt and 50 nt spacers in a luciferase assay screen of tiled guides against the *Ush2a* p.W3947X target (Fig. 1F, G). The 30 nt guides all demonstrated editing rates of less than 10% but editing rates of up to 21% were seen with the 50 nt guides. Of note, a 17% editing rate was achieved with guide 50 nt-28 with the shortest orthologue, Cas13bt3, which is ~600 nt shorter than the dPspCas13(del) construct.

To confirm that on-target editing was due to gRNA-Cas13 interaction recruiting ADAR, rather than over-expressed or endogenous ADAR interacting directly with double stranded RNA, the direct repeat was removed from each of the tested gRNAs. Removal of the direct repeat abolished Firefly restoration to negligible, background levels (Supplementary Fig. 1).

### Comparison of ADAR$_{DD}$ variants and Cas13 truncations

To further refine the system, we compared a range of available ADAR mutants and PspCas13. While the ADAR$_{DD}$(E488Q) domain is commonly used to provide high editing rates, Cox et al. also described the ADAR$_{DD}$(E488Q/T375G) domain, containing a second mutation proposed to decrease the likelihood of off-target editing (Fig. 1H)[12]. The ADAR$_{DD}$(E488Q/T375G) had a significant loss of on-target editing efficiency; however, when compared to the ADAR$_{DD}$(E488Q) using the h*USH2A* targeting guide (50 nt-36) (two-way ANOVA for the effect of ADAR, Tukey's multiple comparison testing, $p = 0.0006$) (Fig. 1I).

The C-terminally truncated dPspCas13b(del) construct is more amenable to packaging within AAV, and we also sought to evaluate its efficiency compared to the full-length dPspCas13b. No loss of efficiency was observed in the C-terminally truncated Cas13b constructs compared to the full-length constructs (Fig. 1I). With the E488Q/T375G ADAR$_{DD}$ no significant differences were observed (Two-way ANOVA for effect of ADAR, $p = 0.99$), while in constructs with the E488Q variant, the C-terminally truncated construct surprisingly had a mean absolute editing rate 8.3% ± 2.3 higher than the full-length construct (two-way ANOVA for effect of Cas, Tukey's multiple comparison testing, $p = 0.027$).

### On- and off-target editing rates by Sanger sequencing

To confirm Firefly restoration was due to A–I editing, PCR amplicons of samples were also subject to Sanger sequencing, demonstrating on-target editing rates of 77% and 86% for *USH2A* and *Ush2a* targets, respectively, with the dPspCas13b(del)-E488Q construct (Fig. 1J). These rates determined by sequencing were higher than those found from the luciferase assay but correlated with rates found with luciferase screening.

Finally, bystander deamination of local adenosines in the target was assessed across each construct (Fig. 1K). The E488Q construct demonstrated off-target editing in both the h*USH2A* (2 sites) and m*Ush2a* targets (8 sites). Off-targeting in the m*Ush2a* transcript outside the gRNA duplex region also occurred in corresponding positions with the non-targeting guide, which did not occur in the h*USH2A* target. This suggests there may be characteristics of the target m*Ush2a* sequence or structure that increase the likelihood of off-target deamination even in the absence of a double-

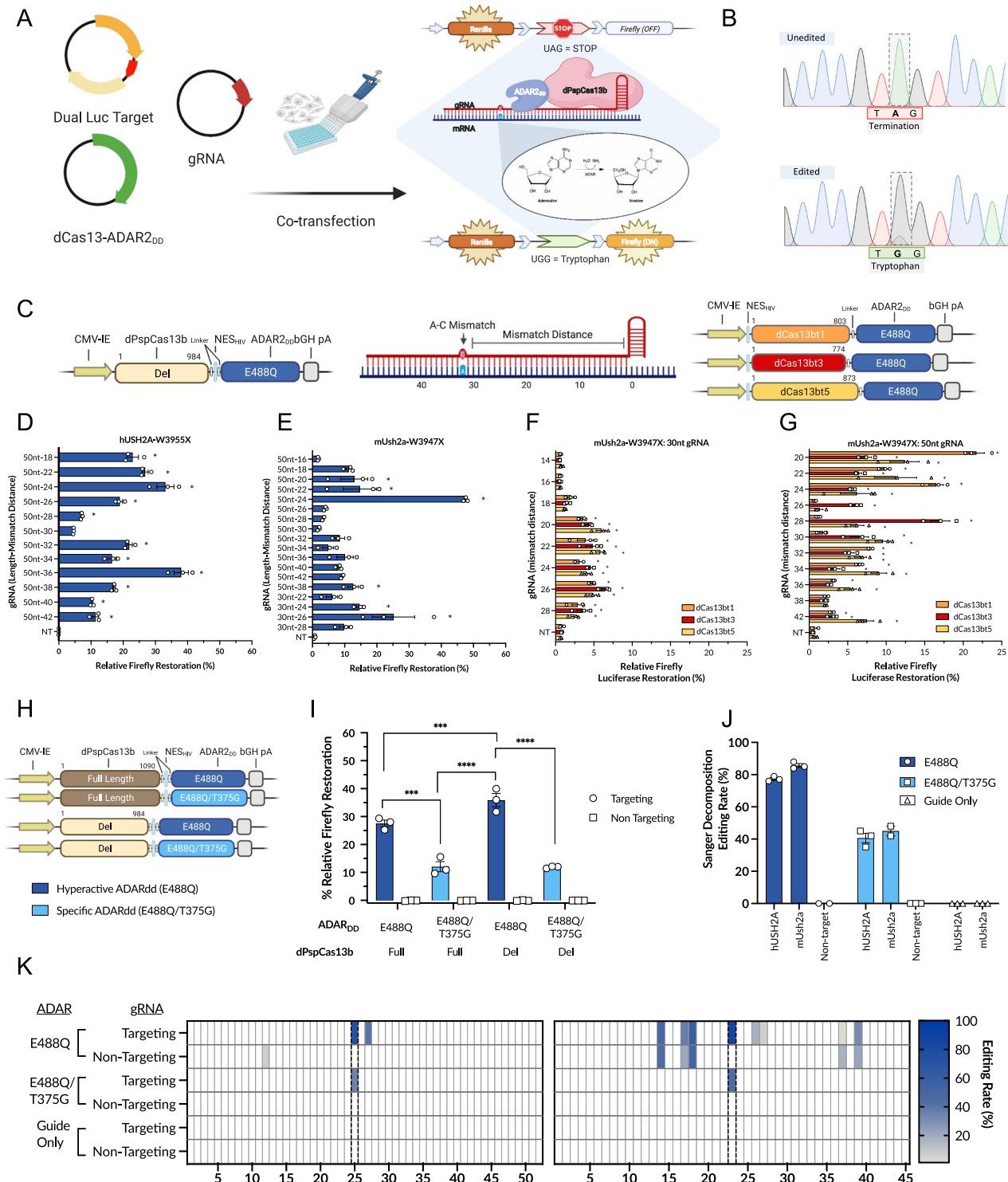

**Fig. 1 | In vitro optimisation of dCas13b-ADAR$_{DD}$ editing of *USH2A*.**
**A** Schematic of triple transfection dual luciferase assay in HEK293T cells for screening of RNA editing. **A–I** editing of a premature stop codon upstream of Firefly results in restoration of Firefly activity. Renilla acts as a normalization control. **B** Representative Sanger sequencing chromatograms following editing. **C** Schematic diagrams of dPspCas13b-ADAR$_{DD}$ and dCas13bt-ADAR$_{DD}$ plasmid constructs, and of gRNA with mismatch distance between (**A–C**) mismatch at target adenosine and gRNA scaffold. **D**, **E** Screening of gRNAs against the *USH2A* p.W3955X and *Ush2a* p.W3947X targets with dPspCas13b(del)-E488Q. One-way ANOVA relative to non-targeting guide, *p* < 0.05. **F**, **G** Screening of 30nt and 50nt gRNAs against

*Ush2a* p.W3947X target with dCas13bt-E488Q constructs. One-way ANOVA relative to non-targeting guide, *p* < 0.05. **H**, **I** Comparison of dPspCas13b-ADAR$_{DD}$ constructs. Two-way ANOVA, Tukey's multiple comparison testing. ***p* < 0.001, *****p* < 0.0001. **J** Quantification of editing rates with dPspCas13b-ADAR$_{DD}$ by decomposition of Sanger chromatograms with MultiEditR. **K** Local bystander deamination of adenosines by dPspCas13b-ADAR$_{DD}$ across the *USH2A* and *Ush2a* target sequences, quantified with MultiEditR. Data all mean ± SEM, *n* = 3. Elements of this figure were created in BioRender. MacLaren (2025) https://BioRender.com/c05j821, l45a427, *x07z781, h76b680, q04x816*.

stranded gRNA-target complex. Comparatively, the E488Q/T375G deaminase mutant was more specific, editing at only the target adenosine in both targets, with no off-target editing with non-targeting guides. No editing was detected in samples transfected with targeting guides only in the absence of Cas13-ADAR.

## Generation of Ush2a$^{W3947X/W3947X}$ mouse for RNA editing

We next sought to develop a mouse model of Usher syndrome containing the Ush2a c.11840 G > A (p.W3947X) premature termination codon for testing RNA editing strategies in vivo. Mice were generated on a C57BL/6 J.Cdh23$^{753A>G}$ background to correct the age-related hearing loss Cdh23 allele present in the wildtype C57BL/6 J strain to eliminate any confounding auditory phenotype due to the Ush2a mutant allele. The target mutation was introduced by CRISPR-Cas9 mediated homology-directed repair using pronuclear injection of zygotes[31] with SpCas9 mRNA, sgRNA and a single-stranded oligonucleotide donor template. F$_0$ offspring were mated with C57BL/6 J.Cdh23$^{753A>G}$ wildtype animals and successful transmission of a single copy of the c.11840 G > A (p.W3947X) mutant allele to the F$_1$ generation was confirmed by Sanger sequencing and a ddPCR copy counting assay (Supplementary Fig. 2). All mice were healthy, fertile and no welfare issues were observed.

Western blot of retinal protein lysates using a C-terminal antibody demonstrated a protein band of expected size in wildtype retinae, with no band detected in Ush2a$^{W3947X/W3947X}$ retinae (Fig. 2A). No truncated bands were observed. In retinal sections, usherin immunoreactivity was not detected at the connecting cilium at the junction between the inner and outer photoreceptor segments in Ush2a$^{W3947X/W3947X}$ 12-week-old homozygous mice, while it was localised correctly in wildtype mice (Fig. 2B)[30].

Although no usherin protein was detected, Ush2a mRNA expression levels from retinal lysates did not demonstrate a difference between wildtype and Ush2a$^{W3947X/W3947X}$ mice (Supplementary Fig. 3). However, when using probes specific to either the wildtype or c.11840 G > A mutation, no wildtype Ush2a was detected in Ush2a$^{W3947X/W3947X}$ and similarly no W3947X transcripts were detected in the wildtype, consistent with the sequencing of RT-PCR products (Supplementary Fig. 3). Together these data suggest that there is minimal non-sense mediated decay of the Us2ha-W3947X transcript.

The retinal phenotype was examined longitudinally every 3 months to 24 months of age. By 24 months, no differences between wildtype and homozygous Ush2a$^{W3947X/W3947X}$ mice were observed in the a-wave and b-wave of scotopic or photopic full-field electroretinogram (Supplementary Fig. 4), or in the thickness of the whole retina, photoreceptor layer or outer segment length assessed by in vivo optical coherence tomography (OCT) imaging (Supplementary Fig. 5).

As Ush2a is also expressed in hair cells of the developing cochlea, immunostaining of cochleas from P4 mice was performed. Usherin staining was absent in W3947X mice, compared to strong staining at the bases of the stereociliary bundle in wild-type mice (Fig. 2C).

To examine the functional effect on hearing, auditory brainstem response (ABR) thresholds to a range of tone frequencies were measured in 9-week-old mice. Like the Ush2a$^{-/-}$ mouse[32], ABR thresholds in Ush2a$^{W3947X/W3947X}$ mice were significantly elevated ($p < 0.0001$, two-way ANOVA), with the effect predominantly observed at higher frequencies (8–24 Hz, $p < 0.05$, Sidak's multiple comparisons). This indicates a likely deficit in auditory nerve response consistent with hearing impairment (Fig. 2D). Observation of the mice did not reveal any evidence of vestibular impairment such as head-bobbing, shaker-waltzer behaviour, or abnormalities in trunk-curl responses or self-righting[33].

These results indicate that mice homozygous for Ush2a c.11840 G > A (p.W3947X) express high levels of the mutant transcript but demonstrate an absence of full-length usherin protein in the retina. This presents a useful model for in vivo testing of gene editing approaches designed to repair the underlying mutation and evaluate restoration of usherin expression.

## Generation of RNA editing transgenes for AAV delivery of dPspCas13b

Constructs were generated to enable delivery of the Cas13 transgenes using AAV. To fit the dPspCas13-E488Q (4.15 kb) within the ~4.7 kb packaging capacity of an AAV vector, small promoter and polyA elements were tested using the luciferase assay.

Compared to the cytomegalovirus (CMV-IE, 588 bp) promoter, the minimal elongation factor-1α (EFS, 212 bp) promoter maintained equivalent editing rates, but the editing rates using the synthetic super core promoter 1 (SCP1, 80 bp)[34,35] were reduced by a mean of 45% (Supplementary Fig. 6A).

For the polyA tail, dual copies of the highly minimal soluble neuropilin-1 (sNRP1, 32 bp)[36] polyadenylation signal only resulted in a slight and non-significant reduction in editing activity compared to when the robust bovine growth hormone (bGH, 225 bp) polyadenylation signal was used (Supplementary Fig. 6B).

For the gRNA promoter, the strong constitutive tRNA$_{GLN}$ (tRNAscan-SE-ID: chr15.trna7, 70 bp) promoter[37,38] was compared to the human U6 promoter (hU6, 249 bp). This resulted in a marked reduction in editing rates and was not used further (Supplementary Fig. 6C).

Based on these data, a dual vector approach was designed for the expression of dPsp-Cas13b. One vector expressed dPsp-Cas13b using an EFS promoter and an sNRP1 polyA tail (Supplementary Fig. 6D) with a total transgene size of 4.81 kb including ITRs (Fig. 3A). The second vector expressed the gRNA under a U6 promoter and encoded GFP to identify successfully transduced cells. These constructs demonstrated equivalent editing efficiency compared to the constructs using the CMV-IE promoter and bGH polyA Western blot and immunocytochemistry demonstrated these plasmids produced the dPspCas13b-ADAR protein at the expected size and correctly localised to the cytoplasm (Supplementary Fig. 6E-F). An additional construct was made, replacing the EFS promoter with the similarly sized rhodopsin kinase (RK, 199 bp) promoter (Fig. 3A)[39], used widely in retinal gene therapy clinical trials[40,41] to enable photoreceptor-specific expression of the construct.

Finally, a third all-in-one AAV vector was designed using the minimal dCas13bt3 orthologue, with the gRNA encoded in the same vector. Of the Cas13b orthologues, dCas13bt3 (also known as Cas13X.1) was chosen for packaging within an all-in-one vector as it is the smallest of the tested orthologues. This allowed space for packaging with an RK promoter, a stronger SV40 polyA terminator, and the U6-gRNA (Fig. 3A) in a single AAV.

## In vivo editing of Ush2a with AAV-delivered Cas13-ADAR

The three designed Cas13-based RNA editing AAV approaches described were then used for in vivo testing in the Ush2a$^{W3947X/W3947X}$ mouse (Fig. 3A). In summary, the first and second approaches included the dual vector dPspCas13b (driven by either the EFS or RK promoter), with the gRNA and GFP on a second vector, the third approach used an all-in-one AAV vector containing both Cas13bt3 (driven by an RK promoter) and the gRNA, but without GFP. All Cas13 proteins were fused to an ADAR(E488Q) deaminase domain as this demonstrated higher rates of on-target editing in culture. AAV8-Y733F vectors were delivered by subretinal injection and analysed as shown in Fig. 3B.

In eyes where a gRNA-GFP vector was injected, successful injection and transduction of both the targeting and non-targeting gRNA vectors were confirmed by widespread retinal GFP expression detected using en-face in vivo imaging with blue autofluorescence (BAF) at 4 weeks post-injection (Fig. 3C). Expression of the transgene from each construct was further confirmed by detection of full-length dCas13-ADAR$_{DD}$ transcripts with RT-PCR (Fig. 3D, Supplementary Fig. 7) performed on RNA from injected retinas, with transcript identity confirmed by Sanger sequencing (Supplementary Fig. 7).

On-target editing efficiency was quantified by targeted deep sequencing with NGS of cDNA from retinas at 4 weeks post-injection (Fig. 3E). The EFS-dPspCas13b vector (dose of 1E + 9gc/eye of Cas13 vector and

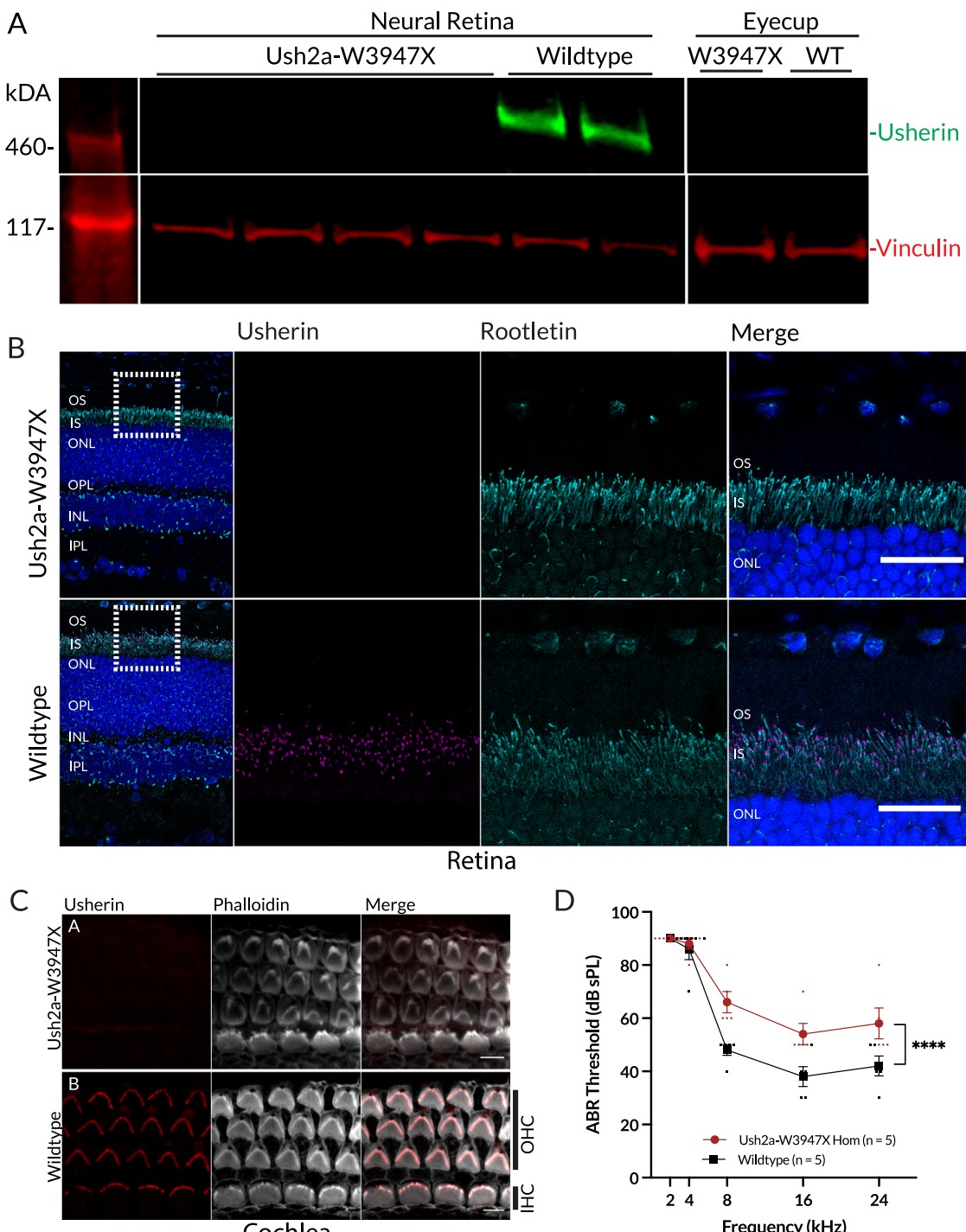

**Fig. 2 | Usherin knockout causes deafness phenotype in *Ush2a*^W3947X/W3947X^ mouse.**
**A** Western blot of protein lysates from the neural retina and eye cup (comprising RPE, sclera and choroid), each lane representing an eye of one animal. No usherin expression is seen in *Ush2a*^W3947X/W3947X^ homozygotes ($n = 4$), with appropriately sized bands (predicted ~590 kDA) in wildtype animals ($n = 2$). Expression is specific to the neural retina. Vinculin used as a loading control. **B** Immunofluorescence images of retinal sections from 11-wk-old animals using a 40× overview (scale = 50 μm) of the full thickness of the retina and a 63× image (scale = 30 μm) of the outer retina (inset). No usherin expression is seen in the *Ush2a*^W3947X/W3947X^ homozygous mouse, while punctate usherin staining is observed in the wild-type

mouse ($n = 4$ per group, representative images). **C** Usherin immunoreactivity (red) is seen at the base of the stereociliary bundle in the inner hair cell layer (IHC) and in the three outer hair cell layers (OHC) in wildtype mice but is absent in *Ush2a*^W3947X/W3947X^ mice. Phalloidin staining of actin filaments is in grey. Scale bars = 5 μm ($n = 3$ per group, representative images). **D** Auditory brainstem responses (ABR) from 9-week-old *Ush2a*^W3947X/W3947X^ homozygous mice and wildtype littermate controls. ABR thresholds are significantly elevated in W3947X mice ($n = 5$ per group, Two-way ANOVA for effect of genotype, $F(1, 40) = 22.92$, $p < 0.0001$) at 8, 16 and 24 Hz frequencies ($p < 0.05$, Sidak's multiple comparison test). Data mean ± SEM.

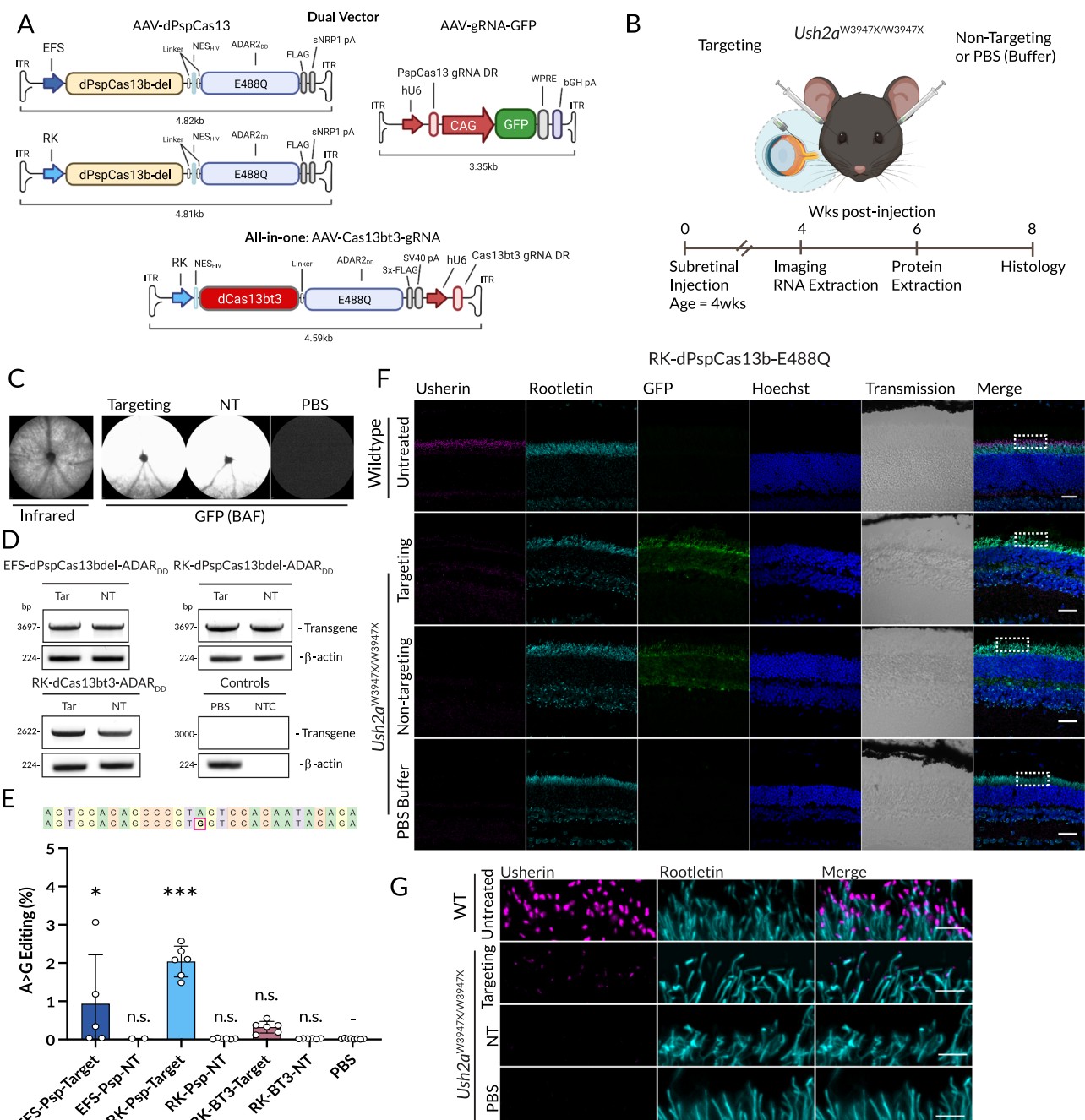

**Fig. 3 | In vivo RNA editing of photoreceptors with AAV-dCas13b-ADAR.**
**A** Construct maps for each of the three injected AAV vectors. The dual vector approach used dPspCas13b-ADAR$_{DD}$(E488Q) driven by either an EFS or RK promoter, together with a gRNA vector also expressing GFP. The all-in-one vector approach included dCas13bt3-ADAR$_{DD}$(E488Q) together with a gRNA.
**B** Experimental outline for in vivo experiments. **C** *En face* in vivo confocal scanning laser ophthalmoscopy (cSLO) retinal imaging. Representative infra-red reflectance image showing optic nerve head centrally. Blue autofluorescence (BAF) imaging demonstrates exogenous GFP expression in injected retinas ($n = 10$ per group), with no expression seen in PBS injected retinas ($n = 5$). **D** Electrophoresis of RT-PCR products of Cas13-ADAR transgenes from injected retinas ($n = 2$ per group). Amplification was not observed in the no-template control (NTC) **E**. On-target A > G editing rates for each construct. One-way ANOVA relative to PBS control with Dunnet's multiple comparisons testing (*$p = 0.0124$, ***$p < 0.0001$). Data

shown as mean ± SEM, $n = 2$–6 as shown. **F, G** Immunohistochemistry of retinal sections from eyes injected with the dual RK-dPspCas13-E488Q vectors scale = 30 µm) ($n = 2$ per group) (**F**) with images focused on the distal inner segment (scale = 5 µm) at areas outlined with boxes (**G**). Endogenous green fluorescent protein (GFP) fluorescence from the gRNA-GFP vector is seen at the injection site in treated animals. Rootletin staining defines the inner segment to the connecting cilium, with Usherin present at the tips. Weak restoration of Usherin is seen in eyes injected with the targeting gRNA in GFP-positive transduced areas of retina. EFS elongation factor-1α short promoter, RK rhodopsin kinase promoter, NT non-target, gRNA guide RNA, DR direct repeat, WPRE woodchuck hepatitis virus posttranscriptional regulatory element, hU6 human U6 promoter, NES nuclear export sequence. Elements of this figure were created in BioRender. MacLaren (2025) https://BioRender.com/ x77a580, n14g404.

1E + 9gc/eye of gRNA vector) demonstrated a mean 0.93% ± 0.57 editing rate in eyes injected with the target vector, however editing was only observed in 2 of 5 eyes. Subsequent investigation using EFS to drive GFP expression following subretinal injection revealed the EFS promoter to have low activity in photoreceptors (Supplementary Fig. 8). Using the photoreceptor-specific RK promoter, the RK-dPspCas13b vector (dose of 1E + 9gc/eye of Cas13 vector and 5E + 8gc/eye of gRNA vector) demonstrated a 2.04% ± 0.16 editing rate. Editing efficiency with the all-in-one RK-dCas13bt3 vector (dose of 1E + 9 gc/eye) was detectable at 0.32% ± 0.06. On-target editing was not detectable in retinas injected with non-targeting gRNAs or those injected with PBS buffer.

Immunofluorescence analysis of retinal sections from injected eyes at 8 weeks post-injection was used to detect usherin restoration. Eyes injected with the dual vector RK-dPspCas13-E488Q vectors, which showed the highest editing rates, demonstrated a low level of usherin restoration in the GFP positive injected area, observed as punctate labelling at the distal inner segment, which was not present in non-targeting or PBS injected animals (Fig. 3F, G). Restoration of usherin staining was not observed in retinas that were injected with either EFS-dPspCas13b or RK-dCas13bt3 vectors (Supplementary Fig. 9).

## Off-target editing of *Ush2a* with AAV-delivered Cas13-ADAR

To explore the off-target conversion of bystander editing of adenosines in the *Ush2a* transcript, the rates of A > G conversion were analysed in vector-injected eyes relative to PBS-injected eyes (Fig. 4A). In eyes injected with the dPspCas13b and the targeting gRNA, off-target editing occurred with a mean ( ± SEM) editing rate of 0.05% ± 0.01 for the EFS-dPspCas13b-E488Q and 0.10% ± 0.03 for RK- dPspCas13b-E488Q (Fig. 4B). For both constructs this was a ratio of approximately 1 off-target edit for 20 on-target edits. Off-target editing with dPspCas13b clustered in the dsRNA region created by gRNA binding. The highest rates of edited sites in injected eyes were positions $A_{35}$ and $A_{36}$ (also identified in the in vitro analysis in Fig. 1K as $A_{26}$ and $A_{27}$) which showed editing rates of up to 0.36% in the RK-driven vector and are predicted to produce missense changes in the protein. A greater rate of editing and a greater number of editing sites were detected in the RK-driven vectors than in the EFS-driven vectors. This may be due to the higher expression leading to higher off-targeting rates in the RK vectors, or the low expression from EFS vectors producing off-target edits below detection sensitivity. Off-target editing at sites distant to the gRNA binding site was seen across both constructs, and additionally, at two sites in eyes injected with the non-targeting RK-dPspCas13b vector. Off-target editing rates in

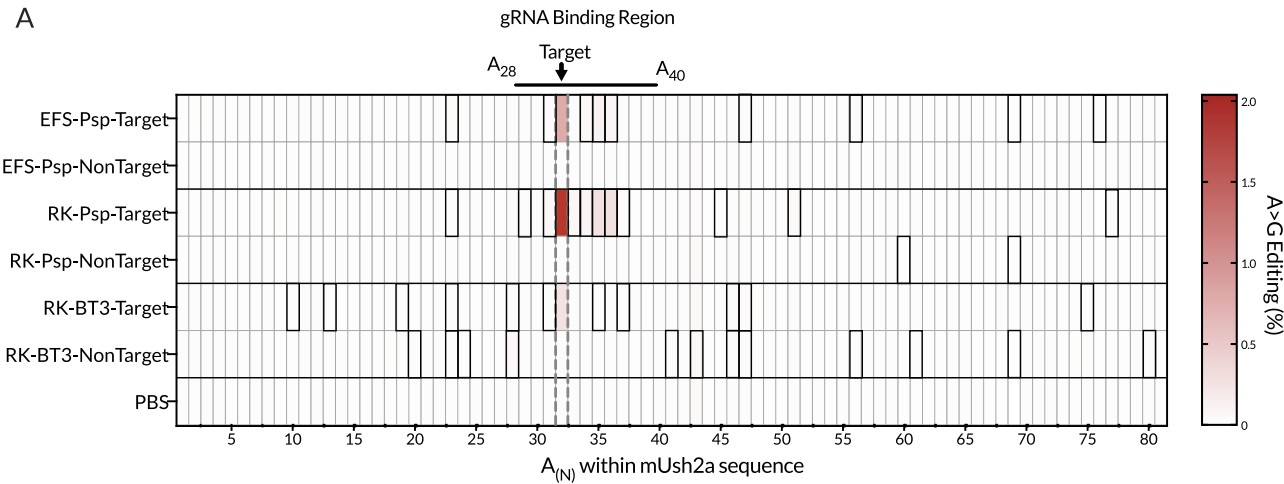

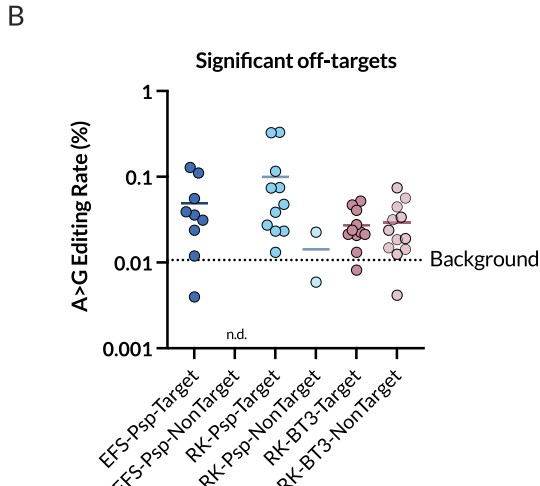

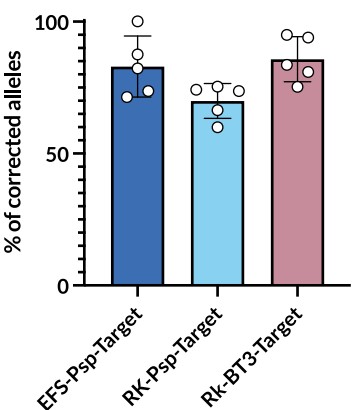

**Fig. 4 | Off-target editing of *Ush2a* with AAV-dCas13b-ADAR. A** Off-target analysis by deep sequencing, demonstrating A > G editing rate at every local adenosine in the Ush2a exon 60 amplicon for each vector (*n* = 6 retinas per vector, PBS *n* = 8). Significant editing sites outlined in black (adjusted *p* < 0.01, PBS-injected eyes as controls). gRNA binding region demonstrated from $A_{28}$ to $A_{40}$. **B** Editing rates of all significant off-target sites plotted on a log scale. Mean background rate of A > G calls in sequencing data in transcripts in PBS injected eyes shown as dotted line. Data = mean ± SEM (*n* = 2–12 as shown in individual points). n.d. not detected. Significant editing sites were determined with Fisher's exact test (two-tailed) with multiple comparison testing using a false discovery rate (FDR) controlled at 5% with the Benjamini-Hochberg method. **C** Proportion of transcripts that were correctly edited (A > G correction at target base) that did not contain any bystander edits at other adenosine sites. Data = mean ± SEM.

the Cas13bt3-injected eyes were lower, but were observed across a greater and more sporadic number of sites outside the gRNA binding region, in both the targeting and non-targeting constructs, demonstrating non-specific deamination of the transcript.

Mapping of the deep-sequencing reads enabled an analysis of the number of transcripts that contained the desired TAG > TGG edit only, without potentially deleterious off-target deamination events, an estimate of the precision of the editing (Fig. 4C). This demonstrated that with each vector, 70–85% of the transcripts that were corrected to the wildtype sequence did not contain any additional off-target A > G edits within the region analysed.

### Effect of AAV-delivered Cas13-ADAR on retinal structure

Optical coherence tomography (OCT) in vivo imaging was performed at four weeks post-injection to search for any early deleterious effects of sub-retinal delivery of AAV-Cas13-ADAR on outer retinal structure. Measurements of photoreceptor layer (PRL) thickness at the injected area (superior) were compared to the uninjected area (inferior) as an internal control, as well as to buffer-injected animals (Fig. 5).

Subretinal injection with a buffer control solution produced a small loss of PRL thickness ($5.9\% \pm 4.4$, $p < 0.05$) in the injected superior retina compared to the inferior retina, as expected from the surgical procedure. Comparing the superior retinal thickness of the injected groups compared to buffer, only the EFS-dPspCas13b-E488Q (targeting) and RK-dPspCas13b (targeting and non-targeting) showed significant thinning ($p < 0.05$).

Comparison of the superior (injected) to inferior (uninjected) retina demonstrated that thinning occurred in response to injection in all groups, and at greater magnitude in those injected with AAV. The EFS-dPspCas13b-E488Q groups were injected with a 1E + 9gc/eye dose of the Cas13 vector and 1E + 9gc/eye of the gRNA vector, with superior thinning of $29\% \pm 6$ ($p < 0.05$) and $20\% \pm 6$ ($p < 0.05$) in the targeting and non-targeting groups, respectively. The RK-dPspCas13b groups were injected with a lower dose of the gRNA vector (1E + 9gc/eye of Cas13 vector and 5E + 8gc/eye of gRNA vector), with superior PRL thickness losses of $24\% \pm 5$ and $26\% \pm 5$ observed in targeting and non-targeting gRNA cohorts, respectively. The RK-Cas13bt3 group was injected with a total dose of 1E + 9gc/eye of the single vector (without GFP), and superior PRL losses of $12 \pm 4\%$ and $17 \pm 5\%$ were seen in the targeting and non-targeting groups, respectively.

Further controls were performed to isolate the cause of retinal thinning. When injected alone without a gRNA-GFP vector, the EFS-dPspCas13b-E488Q vector (1E + 9 gc/eye) resulted in superior PRL thickness loss of $17\% \pm 7$. This was similar to the thinning produced by the gRNA-GFP vector alone at 5E + 8gc/eye ($16\% \pm 5$), but much less than produced by this vector at 1E + 9gc/eye ($41\% \pm 10$). This analysis suggests that the high dose (1E + 9) gRNA-GFP component of the dual vector injections was the most toxic component of the dual vector system.

### Discussion

RNA editing offers a potential therapeutic strategy for correcting editable pathogenic transition variants. In inherited retinal disease, these account for 53% of mutations in genes such as *USH2A* and *ABCA4*, which are both common and not easily amenable to AAV-mediated gene replacement strategies due to their large size[24]. *USH2A* is the second most common gene affected in inherited retinal disease, accounting for approximately 8% of IRDs[22]. In this study, we demonstrate the efficient correction of a common mutation in *USH2A* that causes a pathogenic premature termination codon (c.11864 G > A, p.W3955X) using Cas13-based RNA editors. Furthermore, we show that AAV-mediated in vivo delivery of dPspCas13b-ADAR2dd can be used for the correction of the homologous mutation in the mouse *Ush2a* gene (c.11847 G > A; p.W3947X), the first demonstration of in vivo RNA editing in the retina.

Using luciferase assay screens in cells in culture, we were able to compare Cas13-ADAR systems and guide RNAs. We confirmed previously published data demonstrating the robust on-target activity of the E488Q mutant ADAR and reduced on-target activity of the E488Q/T375G mutation[12]. Encouragingly, the preserved editing activity seen in dPspCas13bdel constructs with the C-terminal 984-1090 deletion confirms that this truncated protein retains its function for on-target editing. While the Cas13bt constructs evaluated here have significant potential due to their minimal size, the activity of these orthologues was markedly lower than PspCas13. Further work using Cas13bt orthologues against other targets would characterise their activity more thoroughly.

We compared guide RNAs over a wide window of mismatch distances. Although a 36nt mismatch distance for the *USH2A*-W3955X was found to be the most active, consistent with initial reports[12], this was not the case for *Ush2a*-W3947X with the most efficient guides being located at a mismatch distance of 24 and 28nt for dPspCas13 and dCas13bt3 respectively. This emphasises the need for screening of many gRNAs for each target. No editing was observed when non-targeting guides or guides without DR scaffold sequences were delivered with Cas13 constructs, supporting the conclusion that editing was gRNA-Cas13 mediated.

In vivo comparison of the different promoters and Cas13 effectors in this study offers insights for the development of therapeutic approaches. Using a dual vector approach with dPspCas13b-ADAR2$_{DD}$, we achieved a 2% correction of the W3947X mutation in *Ush2a* using a photoreceptor-specific RK promoter to drive dPspCas13b-ADAR expression. As this rate is derived from processing of whole retina, editing rates in transduced cells may be up to 3-times higher, as only approximately 30% of the retina is transduced via a single subretinal injection. Further studies to isolate GFP positive populations of cells using cell sorting could confirm this. In comparison, minimal editing was observed using the ubiquitous EFS promoter in vivo despite excellent in vitro efficiency, likely due to the poor photoreceptor activity of the EFS promoter.

Restoration of usherin protein was observed in areas of retina transduced with AAV, with immunohistochemistry demonstrating the protein to be precisely localised to the connecting cilium. Correct localisation provides indirect evidence of restoration of the full-length protein because it needs to be trafficked, folded and then inserted correctly. This requires the c-terminal PDZ1-binding motif which mediates insertion into the connecting cilium membrane[42].

While no RNA editing studies in the retina are reported, other groups have investigated RNA editing in vivo with various approaches. Using a miniaturised Cas13X delivered in a single AAV-PHP vector, Xiao et al. detected a $4.22\% \pm 0.68$ editing efficiency in inner ear hair cells in a model of autosomal dominant hearing loss ($Myo6^{C442Y/C442Y}$), and some recovery of hearing function in heterozygous animals[43]. Other site directed RNA editing systems have also been explored. Using the MS2 bacteriophage coat protein (MCP) editing system fused to an ADAR1$_{DD}$, an AAV8-MCP-ADAR1$_{DD}$(E1008Q) construct delivered with an MS2 gRNA demonstrated 2% on-target efficiency and partial restoration of dystrophin expression in an *mdx* mouse model of muscular dystrophy[44]. Katrekar et al. also employed GluR2 gRNAs which use the Q/R motif in the *GRIA2* transcript, a natural target for the binding of the dsRBD domains of full-length ADAR2[44]. They delivered ADAR2 and ADAR2(E488Q) sequences with GluR2-gRNAs via AAV8 intramuscularly to the *mdx* mouse and systemically to the sparse fur ash ($spf_{ash}$) mouse model of ornithine trans-carbamylase (OTC) deficiency. This resulted in the 0.8–4.7% editing efficiency of a stop codon and splice defect in each model respectively, with resultant rescue of protein expression. In another approach, Yi et al. delivered circularised ADAR recruiting RNAs (LEAPER system) to direct endogenously expressed ADAR to edit targets in the liver of non-human privates, and the central nervous system of mice carrying the humanised $IDUA^{W420X}$ mutation associated with Mucopolysaccharidosis type I, a lysosomal storage disease[45]. Encouragingly, editing rates of up to 50% in the liver and 30% in the brainstem were observed. This approach is tissue dependent and requires adequate endogenous ADAR expression. A consistent feature of all studies is a significant reduction in editing efficiency in vivo compared to in cells in culture.

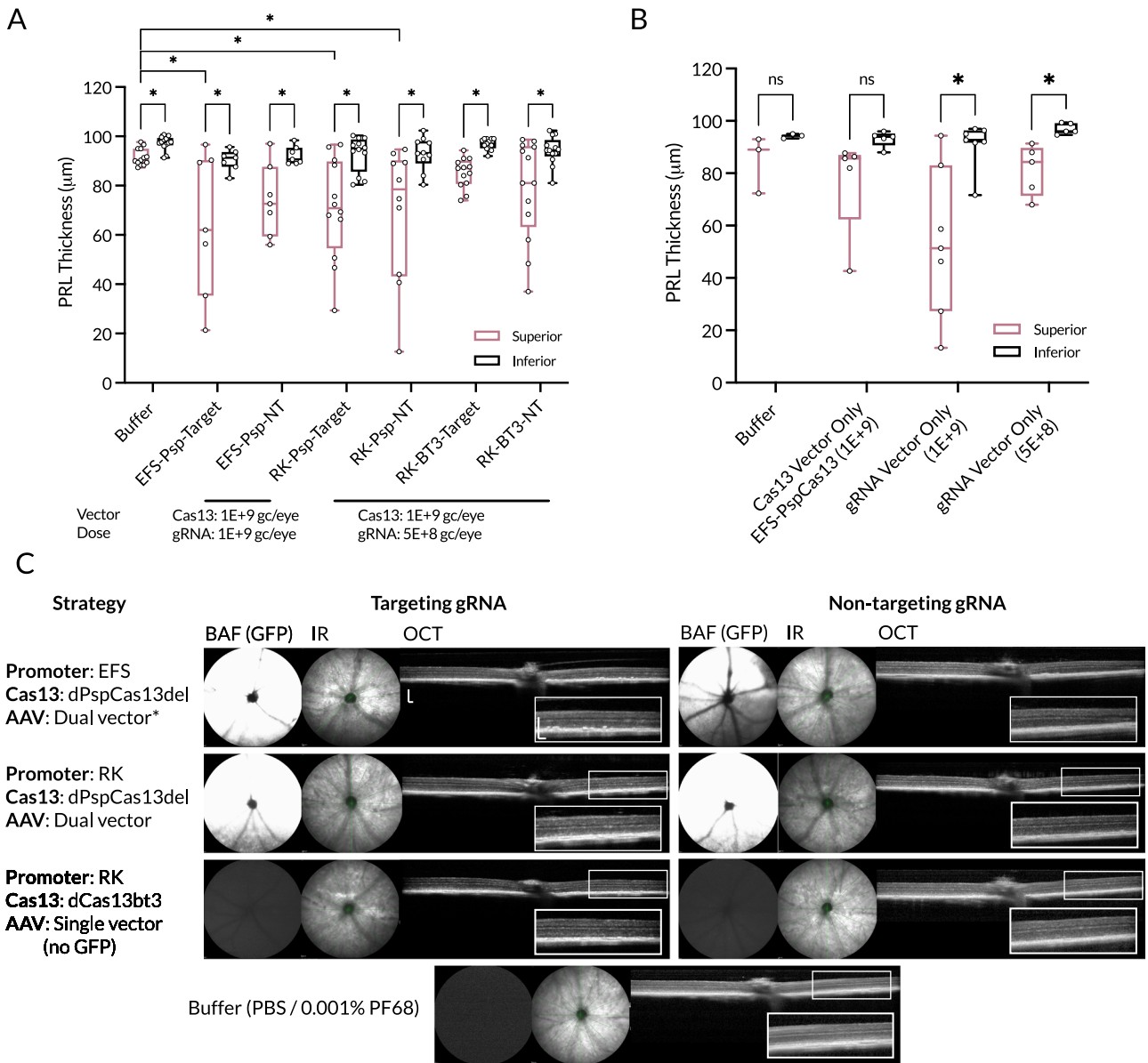

**Fig. 5 | Effect of injection on retinal thickness and structure. A** Photoreceptor layer thickness (PRL) at 4-weeks post subretinal injection measured at the superior retina (at injection site) and inferior retina (opposite injection site). Retinal thinning occurred in all groups following injection, but at a larger magnitude in AAV-injected eyes compared to PBS-injected eyes. Doses administered were $1E + 9$ gc/eye of Cas13 vector and $1E + 9$ gc/eye of gRNA vector for the EFS-Psp group, $1E + 9$ gc/eye of Cas13 vector and $5E + 8$ gc/eye of gRNA vector for the RK-Psp group, and $1E + 9$ gc/eye of single vector Cas13-gRNA for the RK-Cas13bt3 group. Comparison of superior retina to buffer injected groups with one-way ANOVA with Sidak's multiple comparison testing, only showing significant differences. Comparison between superior and inferior retina with paired *t*-tests. n.s. not-significant, *$p < 0.05$, $n = 7–13$ per group as per figure. **B** Photoreceptor layer thickness in single vector only injected eyes (controls) to demonstrate effect of individual vectors and doses on retinal thickness. *Ush2a*$^{W3947X/W3947X}$ mice four-weeks post injection with either AAV-Cas13 only or a targeting AAV-gRNA-GFP vector at a low ($5E + 8$ gc/eye) or high $1E + 9$) gc/eye) dose. Paired *t*-tests (*$p < 0.05$), $n = 3–7$ as shown in figure. **C** CSLO images acquired with blue autofluorescence (BAF) imaging with a bandpass filter (BP), infra-red reflectance (IR) and OCT for each condition. BAF imaging shows robust GFP expression in the superior retina from GFP constructs. The RK-Cas13bt constructs did not contain a GFP expression cassette. In eyes with retinal thinning, this was seen principally in the outer retina, and associated with loss of the hyper-reflective ellipsoid zone and external limiting membrane bands. BAF images acquired at 90 detector sensitivity. Scale bars = 200 μm vertical and horizontal. $n = 7$ to 13, as indicated by individual points in (A). NT non-target.

The overall efficiency of a dual vector approach is restricted by needing two transduction events of the same cell for both the gRNA and Cas13. In contrast to the dual vector approach, the smaller Cas13bt3 protein permitted an all-in-one AAV strategy, though editing rates were also low. Difficulties with obtaining high Cas13bt-ADAR expression and editing with an AAV2-EFS-Cas13bt1-ADAR vector have been noted by other authors[16]. An all-in-one AAV strategy would optimise transduction efficiency and reduce overall AAV dose. Ideally future studies would employ a single vector approach, and this could be achieved using more efficient, short Cas13

effectors. Options include using alternative orthologues or through methods to improve Cas13bt efficiency and expression, such as through codon optimisation.

Off-target deamination events were examined at local adenosines in the target transcript. Deamination events were concentrated in the dsRNA region formed by the gRNA, but also occurred sporadically across the transcript. Sporadic off-target events were more common in the dPspCas13b targeting compared to the non-targeting conditions suggesting that these were increased by ADAR recruitment, however they were also

seen in the non-targeting conditions. Comparatively, the dCas13bt3 generated sporadic off-targets throughout the transcript in both the targeting and non-targeting conditions, suggesting a promiscuous editing profile and low target affinity. Transcriptome-wide off-target analyses were not performed in this study but would be of benefit when on-target editing rates are improved. Studies in cultured cells have noted the hyperactive ADAR-E488Q to have transcriptome wide off-targets[12], and this should be addressed in future studies with optimisation of the effectors to minimise these. It is worth noting that unlike the irreversible off-targets created by DNA editors, RNA editing events are transient and affect only transcripts expressed in the target cells[1]. The risk of unintended editing is mitigated by the knowledge that those transcripts will continue to be degraded. Further approaches to mitigate off-target editing have been reported, such as the incorporation of mismatch wobble bases in gRNA design[46].

Retinal thinning following injection with AAV-Cas13 constructs was also observed on in vivo OCT imaging. A proportion of the loss of photoreceptor layer thickness may be attributed to the gRNA-GFP vector. GFP is very helpful as a marker for transduction, however GFP overexpression with ubiquitous promoter is possibly toxic to photoreceptors at doses at or exceeding 1e + 9 gc/eye[47]. Injection of control vectors such as AAV-GFP and an AAV vector without a functional transgene would help to further understand whether the AAV capsid or GFP contributed to the retinal thinning. A proportion of the retinal thinning may also be due to the dCas13b and/or ADAR2DD(E488Q). Less retinal thinning was seen in the dCas13bt3-injected retinas, which may be related to the lower total dose used to deliver a single rather than dual AAV vector or may be due to the dCas13b used. Katrekar et al. detected toxicity in animals systemically injected with AAV-delivered ADARDD(E488Q), hypothesising that the hyperactive editor was associated with deleterious effects[44]. The immunogenicity of expressing these bacterial-derived proteins in the retina is still to be investigated.

Ongoing protein engineering continues to improve the size, efficiency, and specificity of Cas13 orthologues for RNA editing. A functional miniaturised PspCas13b orthologue with targeted deletions of protein domains designed for AAV packaging has been demonstrated[48], and rational engineering of Cas13bt3 has generated more specific, efficient and smaller variants[19,49]. These add to the armament of RNA editing strategies currently under investigation[7,50].

The Ush2a^W3947X/W3947X mouse was generated with a Cas9-HDR knock-in strategy to contain a mutation analogous to the human USH2A p.W3995X mutation. Molecular characterisation of the model suggests that the homozygous W3947X mutation results in the absence of full-length usherin protein expression in the retina and cochlea. A C-terminally truncated product would be predicted to lack the necessary C-terminal transmembrane domain for correct localisation at the base of the photoreceptor connecting cilium and lack the PDZ-binding domain for interaction with interacting proteins such as with whirlin (WHRN) and adhesion G protein-coupled receptor V1 (ADGVR1, also known as VLGR1). The predicted lack of usherin function in this model is supported by the early hearing loss phenotype.

No retinal degeneration phenotype was observed functionally or structurally in mice aged up to 24 months. In many models of Usher syndrome, protein knockout has been observed to cause a hearing but no retinal phenotype[51], however some element of retinal degeneration has been seen in aged zebrafish[52–54], rabbits[55] and mice[30,56]. Our results contrast to the two other Ush2a mouse models which have demonstrated a mild phenotype with aging. For example, the Ush2a^-/- knockout model demonstrated losses of retinal function measured by optomotor testing and photoreceptor outer segment shortening at 12 months of age with significant degeneration and loss of ERG amplitudes by 20 months[30,57]. The recently described Ush2a^delG/delG humanised knock-in model contains a humanised c.2290delG mutation to simulate the common human c.2299delG mutation[56]. It is not clear, however, whether this mutation is solely responsible for the retinal degeneration phenotype described in the model. Early loss of retinal function was seen at 6 months measured by optomotor function and outer retinal

thinning and losses in scotopic a- and b-wave ERG amplitudes were observed from 12 months. This knock-in has a 20 amino acid extension immediately after the c.2290 deletion followed by a triple (3x) FLAG-tag, and an IRES sequence to enable GFP expression from the Usherin promotor. The mouse expressed a truncated usherin product and mislocalisation of usherin, interacting proteins, and rhodopsin. Accumulation of usherin within the endoplasmic reticulum points to the possibility of the truncated protein causing endoplasmic reticulum stress and degeneration through an unfolded protein response, causing both a loss-of and gain-of function affect[56]. This may explain the earlier and more significant degeneration in the Ush2a^delG/delG model, and it is unclear whether this recapitulates mechanisms of degeneration in humans.

A notable difference is that the Ush2a^W3947X/W3947X mutation lies distally in exon 60 in isoform B of the Ush2a transcript. This contrasts to other models where the mutation lies in the shared proximal sequence of isoform A and isoform B: the Ush2a^-/- mice have an exon 5 knockout while the Ush2a^delG/delG and Ush2a^-/- rabbit model both cause frameshift mutations in exon 12. Further characterisation of the Ush2a^W3947X/W3947X model needs to be undertaken to understand the potential molecular mechanisms for the difference observed between the retinal phenotype of these models.

USH2A is an excellent therapeutic target for a base editing approach. It is a very large and stable structural protein expressed in the photoreceptor cilium. Restoration of a similar large cilium protein CEP290 via RNA editing with antisense oligonucleotides demonstrates restored ciliogenesis in pre-clinical models[58], and durable visual improvements in patients following intravitreal administration[59]. This supports the notion that whilst the corrected RNA half-life might be low, large structural proteins may last for some time in the cilium, and may have a far more durable response than the RNA itself. Only a small restoration of expression may be required to enable gradual accumulated restoration of the protein. Further longitudinal studies will be required to determine how long the expressed vector continues to edit the target, and the half-life of usherin following repair. Consistent with the autosomal recessive inheritance of USH2A-related disease, mice heterozygous for a pathogenic Ush2a allele do not display any phenotype, suggesting the expression of some functional usherin is sufficient to prevent degeneration[56]. While the level of correction required to ameliorate the clinical USH2A phenotype is unknown, in choroideremia, another IRD, we have shown that the presence of only approximately 5% of wild-type transcript was required to slow disease progression in patients[60]. The lack of retinal phenotype in this mouse reduces the utility of this model for evaluating the functional effect of gene therapies, however it remains useful for testing the in vivo efficiency of gene editing therapeutics and for restoration of protein expression.

In conclusion, we have demonstrated the use of a CRISPR-Cas13 RNA editing approach for the correction of a common pathogenic USH2A mutation, and that RNA editing is capable of in vivo correction of photoreceptor transcripts and restoration of usherin protein expression in a novel mouse model. Building on these findings, studies to improve the efficiency and specificity of these tools will improve the use of Cas13-RNA editors as a therapeutic approach for inherited retinal disease.

## Materials and Methods
### Plasmids and cloning
Plasmids obtained for use in this study are detailed in Supplementary Table 1. Oligonucleotide sequences are detailed in Supplementary Table 2. Plasmid identity was confirmed with Sanger sequencing. Cloning oligos and double-stranded fragments were synthesised (IDT, USA). Plasmids were cloned using restriction enzyme cloning or using multi-fragment assembly with the NEBuilder HiFi DNA assembly kit (NEB, USA) following the manufacturer's instructions.

Plasmids encoding variants of deactivated Cas13b derived from Prevotella Sp. (dPspCas13b) with an HIV nuclear export signal (NES_HIV) fused via a linker to the human ADAR2 deaminase domain (hADAR2_DD), the minimal deactivated Cas13bt orthologues and associated guide RNA constructs were a gift from Feng Zhang[12,16].

Dual luciferase constructs for RNA editing reporting were cloned using the pSGDlucV3.0 plasmid (gift from John Atkins)[61,62]. Target cassettes containing either an in-frame positive control sequence or a G > A mutation of interest and the 200 bp region surrounding the *hUSH2A*-W3955X and *mUsh2a*-W3947X mutations were synthesised as double-stranded DNA fragments and cloned into the 5′ PspXi and 3′ BglII flanking restriction sites. From a single transcript, Renilla was expressed as a normalization control, a downstream 'target cassette' containing the premature termination mutation of interest was cloned between flanking F2A sequences, while downstream of the target cassette, Firefly was encoded as a reporter of editing activity. At baseline, the upstream termination codon prevents Firefly expression, but repair of the premature stop codon to a non-stop amino acid codon allows translation to proceed and Firefly is expressed.

To create plasmids for AAV production, prepared fragments were cloned using multi-fragment assembly into an AAV plasmid backbone produced by digestion of the pX601 vector (gift from Feng Zhang) with NotI and XbaI restriction enzymes to excise the CMV-SaCas9-gRNA transgene, leaving flanking AAV2 ITRs. Transgenes were cloned within these ITRs.

Plasmids for testing of short promoter sequences were sub-cloned. The CMV-IE promoter was excised from the dPspCas13bdel-ADAR2$_{DD}$(E488Q) vector at the SpeI and HindIII sites. The SCP1 and EFS promoters were synthesised as double-stranded DNA fragments (IDT) and cloned into the linearised backbone. For cloning the sNRP1 polyA sequence, a double-stranded fragment (IDT) encoding a dual sNRP1 was cloned downstream of dPspCas13bdel-ADAR2$_{DD}$(E488Q). The tRNA(GLN) promoter was synthesised and cloned into the pC0043-pspCas13 empty gRNA cloning backbone to replace the U6 promoter using in vivo assembly[63].

## Dual luciferase assay

The dual luciferase assay was conducted in cultured HEK293T cells (American Type Culture Collection ATCC) seeded in Corning white-walled 96-well adherent plates (Sigma-Aldrich) in phenol-red-free Dulbecco's Modified Eagle Medium (DMEM) (Thermo Fisher Scientific, UK), supplemented with L-glutamine (2 mM) (Sigma-Aldrich, UK), sodium pyruvate (1 mM) (Thermo Fisher Scientific), 10% heat-inactivated foetal bovine serum (HI-FBS) (Thermo Fisher Scientific) and 1% penicillin-streptomycin (Sigma-Aldrich). Cells were transfected with Transit-LT1 (Mirus Bio) at 24 h post-seeding with 50 ng of dual luciferase reporter, 75 ng of RNA editing effector and 150 ng of gRNA. Positive and negative control wells were included on all plates, transfected with identical quantities of luciferase plasmid, RNA editing effector and non-targeting gRNA plasmids as in the experimental plates. All transfected plasmid preparations were made with endotoxin-free mini- (Zymo Research, Germany) or maxi-prep kits (Qiagen).

In the dual luciferase plasmid, a single SV40 promoter drives the production of a transcript encoding Renilla as a normalisation control, a 'target cassette' containing the mutation of interest with flanking F2A sequences, Firefly as a reporter of editing activity and a SV40 PolyA site from 5′ to 3′. The two F2A sequences enable co-translational excision of the target cassette peptide, encoding Renilla and Firefly proteins independent of the identity of the intervening target cassette sequence. At baseline, Firefly is turned off as the presence of the termination codon stalls translation after the production of Renilla but before the production of Firefly. Repair of the premature stop codon to a non-stop amino acid codon allows translation to proceed and Firefly is turned on.

The Dual-Glo luciferase assay kit (Promega, Madison, WI, USA) was used to detect luciferase expression at 48 h post-transfection according to the manufacturer's instructions. Briefly, plates were allowed to come to room temperature, and 50 μL of media was removed. For cell lysis and detection of Firefly activity, 60 μL of Dual-Glo luciferase reagent containing Firefly luciferase substrate was added to each well and left to incubate at room temperature for 15 min. Luminescence was recorded using a FLUOstar Omega microplate reader (BMG Labtech, Germany). For Renilla detection, 60 μL of Stop & Glo reagent was added to quench Firefly activity and provide Renilla luciferase substrate, left to incubate for 15 min and luminescence recorded.

Mean background Firefly and Renilla luminescence was calculated from transfection-reagent-only wells and subtracted from all raw values. The editing efficiency is calculated from a ratio of Firefly to Renilla luciferase activity, determined for each well and normalised between the positive control and negative control ratios. All assays were performed in technical duplicates, and the experimental sample ratio is the mean of the duplicates for that condition.

## AAV production

Vectors with an AAV8-Y733F capsid were produced using a triple transfection method[64]. HEK293T cells seeded into two CellBIND Surface HYPERflasks (Corning, USA) per vector were transfected with the cationic polymer polyethylenimine (PEI) and 500 μg total DNA composed of the selected transgene plasmid (pTransgene), pRepCap and pHelper. After three days, cells were harvested and lysed. AAV was isolated by ultracentrifugation with an iodixanol gradient, purified using an Amicon Ultra-15 100 K filter (MerckMillipore, UK) and eluted in phosphate-buffered saline (PBS).

AAV preparation purity was analysed with SDS-PAGE, and endotoxin testing was performed using the Pierce *Limulus* Amebocyte Lysate (LAL) Chromogenic Endotoxin Quantitiation Kit (Thermo Fisher Scientific) to ensure endotoxin levels <1EU/mL. The concentration of DNA-se resistant viral genomes was determined using SYBR Green qPCR with transgene-specific primers. For injection, vector preparations were diluted in PBS + 0.001% Pluronic F68 (LifeTechnologies, UK), which was also used as the vehicle injection for sham injected eyes.

## Animal husbandry

All animal work was performed in accordance with the Animals (Scientific Procedures) Act 1986, UK, with the Association for Research in Vision & Ophthalmology (ARVO) statements on the care and use of animals in ophthalmic research. Mice were housed in individually ventilated cages with a 12-h light/dark cycle and food and water available ad libitum. All procedures were evaluated and approved by the local Animal Research Ethics Committee of the University of Oxford, carried out under a Home Office Project Licence. We have complied with all relevant ethical regulations for animal use. Approximately equal proportions of male and female mice were used for each experimental cohort.

For procedures requiring general anaesthesia, intraperitoneal injection of a cocktail of ketamine (Narketan, Vetoquinol, Magny-Vernois, France, final concentration 8 mg/mL) and xylazine (Rompun, Bayer, Germany, final concentration 1 mg/mL) in normal saline at a dose of 0.1 mL/10 g bodyweight was used. Anesthesia was reversed with intraperitoneal injection of atipamezole (Antisedan, Zoetis, NJ, USA, final concentration 0.2 mg/mL) in normal saline at a dose of 0.1 mL/10 g bodyweight. Pupils were dilated using tropicamide 1% and phenylephrine 2.5% topical eye drops (Bausch+ Lomb, Canada).

## Generation of *Ush2a*$^{W3947X/W3947X}$ mouse model

As C57BL/6 J strains carry a fixed hypomorphic Cadherin-23 allele (*Cdh23*$^{ahl(753 G > A)}$) with a high-frequency age-related hearing loss phenotype, the C57BL/6 J background with a repaired *Cdh23* allele (*Cdh23*$^{753A>G}$)[31] was used for mouse model generation and as wildtype breeders.

The *Ush2a*-p.W3947X (c.11840 G > A) mutation was inserted using a homology directed repair knock-in approach using a single-stranded oligonucleotide DNA (ssDNA) template with flanking homology arms (Mary Lyon Centre, MRC Harwell, Oxfordshire, UK)[31]. A gRNA was designed to target a SpyCas9 PAM site at the target tryptophan codon. Successful insertion of the TAG codon at this locus abolished the PAM site to prevent re-cleavage of the successfully inserted donor sequence. SpyCas9 protein, sgRNAs and ssODNs were diluted and mixed in electroporation buffer (Gibco Opti-MEM I Reduced Serum Media, Thermo Fisher Scientific) to

working concentrations of 650 ng/μl, 130 ng/μl and 400 ng/μl, respectively. Embryos were electroporated using the following conditions: 30 V, 3 ms pulse length, 100 ms pulse interval, 12 pulses and then implanted in pseudo-pregnant CD1 females. F0 offspring were mated with wild-type animals to produce founder heterozygote animals. F1 offspring were back-crossed to wildtype, and heterozygous inter-crosses of the F2 generation produced homozygous W3947X and wildtype littermates.

## Genotyping and copy-counting assay

Genotyping was performed using a multiplexed qPCR-based allelic discrimination assay using primers common to both alleles and fluorescence-conjugated probes specific to either the wildtype (FAM-labelled) or mutant (TET-labelled) allele. DNA was extracted from ear clips using Applied Biosystems TaqMan Sample-to-SNP Kit (Thermo Fisher Scientific). qPCR was performed with an Applied Biosystems 7500 thermocycler (Thermo-Fisher Scientific) with 10 μL reactions containing ABI GTX TaqMan Mastermix (5 μL), Probe-Primer Assay (2 μL) containing WT and mutant probes (5 μM) and forward and reverse primers (15 μM), ddH2O (0.5 μL) and DNA (1:10 dilution of ABI Sample to SNP preparation, 2.5 μL). Genotypes were confirmed where appropriate with Sanger sequencing of amplicons of the target exon.

Digital droplet PCR (ddPCR) copy counting assays of the knock-in template were performed using Dot1l as the normalisation control. Assays were performed with a QX200 ddPCR system (Bio-Rad Laboratories) with 20 μL reactions of 2 μL crude DNA lysate, 1X ddPCR Supermix (Bio-Rad Laboratories), 225 nM of each primer (two primers per assay) and 50 nM of probes. Results were analysed using Quantasoft software.

## Auditory brainstem responses

Anaesthetised mice were placed in anechoic chamber on a heated pad and monitored remotely via video feed. Sound stimuli were presented using a free-field loudspeaker (VISATON FR10) located 8 cm in front of the animal and connected to an amplifier (RA15, Alesis, USA). Auditory stimuli were generated using an RP2.1 Enhanced Real-time processor (Tucker Davis Technologies (TDT), USA) with a sampling frequency of 100 kHz connected to a TDT PA5 programmable attenuator. The speaker was calibrated using SigCalRP TDT calibration software to generate compensation filters ensuring stable intensity levels for a frequency range from 250 to 30,000 Hz. Click stimuli were presented at a rate of 17/sec at intensities from 90 to 30 dB sound pressure level (SPL) and pure tones at increasing frequencies were presented at a rate of 21/sec at intensities from 90 to 20 dB SPL, attenuating in 10 dB steps. Signals were recorded from two active subcutaneous electrodes, placed close to the left and right auditory bullae, respectively, and referenced to an electrode placed at the vertex of the skull. A ground electrode was placed on the back of the animal. The signal was routed to a preamplifier (Medusa TDT RA16PA) via a low impedance headstage (TDT RA4LI) and recorded by an RZ2 Bioamp Processor controlled by BioSigRP software (TDT). Average responses from 1024 stimuli were recorded with BioSigRZ software (TDT) with a recording time of 10 m, a gain of 20 and a 0.3–3000 kHz bandpass filter. Waveforms were analysed by a blinded assessor with thresholds determined as the minimum sound pressure level (SPL) that wave 1 of the ABR, derived from the auditory nerve, could be observed[33,65].

## Retinal imaging

Retinal imaging was performed using a Spectralis (Heidelberg Engineering, Germany) device fitted with a 55° lens[66]. Mice were anaesthetized, pupils dilated and a custom made contact lens applied (Cantor and Nissel, UK) with hypromellose 0.5%[66]. *En face* blue (BAF) and near-infra-red (NIR) fundus autofluorescence images were acquired with the confocal scanning laser ophthalmoscope (cSLO) using 488 and 790 nm excitation light, respectively. For acquisition of images for in vivo EGFP quantification, BAF mode was used with a 500–700 nm bandpass filter to eliminate autofluorescence outside the EGFP emission spectrum[67]. Images were collected at sensitivities in descending steps from 100 to 50%.

Spectral-domain optical coherence tomography (OCT) images with simultaneous recording of a near-infra-red (820 nm) reflectance image were acquired using a standard protocol to capture four line scans oriented radially extending from the optic nerve head as previously described[68]. Photoreceptor layer (PRL) thickness was measured at a fixed 3000 μm distance from the optic disc defined by an overlaid Early Treatment of Diabetic Retinopathy Treatment Study (EDTRS) grid centred on the optic disc in the HEYEX software. PRL thickness was measured using software callipers between the inferior aspect of the hyper-reflective outer plexiform band to the superior aspect of the hyper-reflective RPE band.

## Electroretinography

Active contact lens electrodes were created by attaching silver-coated nylon thread Dawson, Trick & Litzkow (DTL) Plus electrodes (Diagnosys Ltd., UK) to custom contact lenses created from achromatic Aclar (Honeywell International, USA). Mice were dark-adapted overnight for a minimum of 12 h in a light-proof cabinet. Mice were anaesthetised and eyes dilated. Topical hypromellose 1% (Alcon, Switzerland) coupling solution was applied bilaterally and an active contact lens electrode placed centrally on each cornea. Active electrode impedance was maintained between 5 and 10 kΩ. Procedures were performed in a light-proof dark room with dim-red illumination. Mice were placed in a heat-controlled Faraday cage containing a Ganzfield stimulator dome (Colordome Electroretinography Machine, Diagnosys).

Electroretinography (ERG) was performed using a custom stimulus protocol controlled by Espion v6 software (Diagnosys Ltd.) detailed in Supplementary Table 3[68,69]. This comprised a dark-adapted stimulus response protocol of single flashes increasing in log steps from $10^{-6}$ cd.s/m² to 25 cd.s/m² and flicker stimuli. Light-adapted single flash and flicker responses were also recorded. Amplitudes of a- and b-waves were measured manually using Espion v6 software[70]. Flicker stimuli amplitudes were measured from the peak to the trough at approximately 200 ms post-stimuli onset. For all recordings, a 0–500 Hz bandpass filter was used with line filtering, an amplifier gain of 8, a 4000 Hz sampling frequency and 600 ms sweep length.

## Subretinal injection

Corneal paracentesis was performed with a 33 G needle (TSK laboratory, Netherlands) in the mid-peripheral cornea to lower the intraocular pressure. Injections were performed with a 35 G bevelled NanoFil needle mounted on a Nanofil 10 μl syringe. A total volume of 1.5 μl was delivered into the superior subretinal spaced, by the creation of a bullous superior hemi-retinal bleb under the operating microscope. Any complications preventing the creation of a successful bleb, such as haemorrhage, extraocular reflux of injected substance, or inadvertent retinotomy were recorded at the time of surgery, and affected eyes were excluded from analysis.

## RNA extraction and reverse transcription

For RNA extraction from cells to obtain mRNA, the RNeasy Plus Mini kit (Qiagen) was used. Cells were lysed using the RLT Plus buffer supplemented with 10 μl β-mercaptoethanol per mL and homogenised by vortexing. DNA was removed from lysates with both supplied DNA eliminator columns and by on-column DNAse digestion (Qiagen).

For retinal tissue, retinas were dissected and snap frozen on dry ice and stored at −80 °C. RNA extraction was performed with the phenol-chloroform based (TRIzol) miRNeasy Mini kit (Qiagen). Thawed retinae were homogenised and disrupted using a handheld homogeniser and pestle and then RNA was isolated and eluted with RNAse-free water.

A standardised amount (500 ng) of RNA was reverse transcribed to complementary DNA (cDNA) with OligoDT$_{20}$ using the Superscript III first strand synthesis system (Thermo Fisher Scientific). Samples were subsequently treated with RNAse H solution and then the cDNA was purified with a PCR Purification Kit (Qiagen). Negative controls were run in parallel where no reverse transcriptase was added to the reaction.

## RNA sequencing and analysis

RNA was extracted from cultured cells harvested 48 h following transfection. Target regions were amplified from cDNA using high fidelity KOD Hotstart Mastermix (Merck, UK) and 5 μL of cDNA template. Successful amplification of the product was confirmed by gel electrophoresis and then subject to bidirectional Sanger sequencing. RNA editing rates determined using MultiEditR software to perform decomposition of base peaks from Sanger sequencing chromatograms[71–73]. A chromatogram from cells transfected with the target plasmid only was used as the reference sequence. Chromatograms were inspected for quality prior to analysis and only trimmed chromatograms with >80% high-quality base calls (defined as Phred Quality > 40, >99.99% base call accuracy) were included for analysis. The following settings were used for analysis of A > G editing sites: motif of interest (A), WT Base (A), Edited Base (G), Phred Cut-off (0.001, equivalent to Phred Quality of 30), $p$-value cut-off (0.0001), multiple comparison correction (yes).

RNA extracted from neural retina was subjected to targeted deep sequencing. *Ush2a* exon 60 was PCR amplified from cDNA with Phusion High Fidelity polymerase using exon-junction spanning primers with partial Illumina adaptors. Products were visualised on an agarose gel to confirm the amplification of a single product and column purified (Qiagen). Amplicon purity was analysed with spectrophotometry and concentrations were measured with a Qubit high-sensitivity dsDNA kit (both Thermo Fisher Scientific). Amplicons were sent for library preparation and Illumina sequencing at GENEWIZ (Hope End, United Kingdom). Sequencing was performed with a $2 \times 250$ bp paired-end configuration, with adaptor-trimmed Raw Fastq data provided for analysis. Fastq files were aligned, filtered, and analysed with CRISPResso2 (v2.1.3) to calculate editing efficiency from counts of A and G nucleotides across all adenosines in the amplicon[74]. Reads were filtered with a minimum average read quality >30 and minimum single base quality >20, with 15,000–30,000 aligned reads analysed per sample post-filtering. Fisher's exact test (two-tailed) was performed (fisher.test function) with multiple comparison testing using a false discovery rate (FDR) controlled at 5% with the Benjamini-Hochberg method (p.adjust function) using base R (v4.0.0), comparing experimental to PBS injected eyes on pooled count data from all replicates for each condition, with editing sites with an adjusted $p < 0.01$ deemed significant[75].

## RT-PCR of expressed transgenes

For analysis of potential truncation of the expressed transgene in neural retina, Phusion High Fidelity DNA Polymerase (NEB) was used to amplify target amplicons of the transgene expressed in cDNA of AAV-transduced retinas. Combinations of primer pairs targeting the 5' and 3' region of the transcript were used, and product length analysed using agarose gel electrophoresis.

## Protein extraction and western blot

Radio-immunoprecipitation assay (RIPA) buffer (Sigma-Aldrich) was mixed with cOmplete mini protease inhibitor (Sigma-Aldrich) and used to resuspend and lyse the cell pellet or retinal tissue. Cells were disrupted with a handheld rotor, left on ice for 30 min, centrifuged for 10 min at 10,000 $g$ at 4 °C, and the supernatant isolated.

Lysate total protein was quantified using the Pierce bicinchoninic acid (BCA) protein assay kit (Thermo Fisher Scientific) in accordance with the manufacturer's instructions with absorbance readings measured at 562 nm with a plate reader (iMark, Bio-Rad Laboratories). Protein concentration was determined using a standard curve derived from diluted bovine serum albumin (BSA) standards (25–2000 μg/mL.). Protein lysates were prepared to contain an equal mass of protein made up to equal volumes with RIPA buffer, and combined with 5X reducing sample buffer (National Diagnostics, USA) and left at room temperature for 15 min.

Samples were loaded to wells adjacent to 5 μL protein size ladder (BLUeye, GeneFlow) in pre-cast Tris-Glycine gels (Criterion TGX, Bio-Rad Laboratories) submerged in Tris-Glycine Sodium-Dodecyl-Sulphate PAGE(SDS-PAGE) running buffer (National Diagnostics, USA) and subject to electrophoresis. Protein was transferred onto methanol activated low fluorescence polyvinylidene difluoride (PVDF) membranes (Bio-Rad Laboratories) using the semi-dry Trans-Blot Turbo transfer system (Bio-Rad Laboratories) set to the 'Mixed Weight' (2.5 A/25 V, 7 min) programme.

For successful blotting of the large usherin protein from retina samples, 60–80 μg of protein was loaded into NuPAGE 3–8%, Tris-Acetate gels in Tris-Acetate SDS Running Buffer and run with a HiMarker ladder (all Thermo Fisher Scientific). Proteins were transferred overnight at 4 °C in 1X Towbin Buffer with 10% methanol and 0.05% SDS to low fluorescence PVDF membranes using the Criterion system (Bio-Rad Laboratories) set to a current of 70 mA.

Membranes were blocked in 10% skim milk for two hours, before overnight incubation at 4 °C with primary antibodies. Following washing in TBS with 0.01% Tween-20 (Sigma-Aldrich) (TBS-T), membranes were incubated in secondary antibody solution containing appropriate fluorescent labelled antibodies (IRDye, LI-COR Biosciences) for 1 h at room temperature protected from light. Membranes were washed in TBS-T before imaging. Membranes were imaged using the Odyssey Fc imaging system (LI-COR Biosciences) using 700 nm and 800 nm detection channels.

## Retinal immunohistochemistry

Dissected eye cups were fixed in cold 4% paraformaldehyde (PFA) on ice for 30 minutes and then washed briefly in PBS. The eyecup was cryoprotected using a sucrose gradient and then embedded and frozen in optimal cutting temperature (OCT) compound. Frozen sections 16–18 μm thickness were cut and mounted on slides.

Slides were washed with a standard wash step by immersion in 0.01 M PBS three times for five minutes per wash. Sections were blocked and permeabilised in 400 μL 0.01 M PBS with 5–10% serum (Normal Goat Serum, NGS; or Normal Donkey Serum, NDS), 1% bovine serum albumin (BSA) and 0.2% Triton X for 20 min at room temperature followed by a wash step. Slides were incubated overnight at 4 °C in primary antibody solution diluted in 0.01 M PBS with 1% serum, 1% BSA and 0.1% Triton X-100, and then washed. Slides were incubated at room temperature for two hours in secondary antibody diluted in 0.01 M PBS with 1% serum, 1% BSA and 0.1% Triton X-100, and then washed. Following Hoescht staining, slides were mounted in ProLong Dimaond Antifade mounting medium (Thermo Fisher Scientific). Antibodies used are detailed in Supplementary Table 4.

## Cochlea immunohistochemistry

Cochlea were dissected and processed from P4 mice[76,77]. Following dissection, the cartilaginous temporal bone was removed to expose the cochlea, and then the cochlea was bluntly dissected and removed. The otic capsule was gently removed to expose the organ of Corti, and then the spiral ligament was carefully dissected away. The cochlea was then divided into apical, middle, and basal portions. The tectorial membrane was then removed from each portion.

Immunolabelling of wholemounts was performed in a 48-well plate. Following fixation in 4% PFA on ice for 30 min, cochleas were washed and then permeabilised in 0.5% Triton X-100 for 15 min and incubated in blocking solution (8% NGS/5% BSA/0.5% Triton X-100) for 1 h at room temperature. Cochleas were incubated in primary antibody solution at 4 °C overnight (anti-USH2A in 5% NGS/0.1% Triton X-100), then washed and labelled with a secondary solution (5% NGS/0.1% Triton X-100) containing Alexa Fluor 488-conjugated phalloidin and goat anti-rabbit Alexa Fluor 568 for 90 min at room temperature.

## Statistics and reproducibility

Statistics and group sizes are described for each figure. Data were assessed for normality and analysed using standard statistical tests for normal and non-normal data as described for each figure using GraphPad Prism (version 10) software. Comparisons between groups were made by assessing effect sizes, and significance determined with a $p$ value of <0.05 where

appropriate. Each replicate represents an individual measurement from a separate sample.

## Reporting summary

Further information on research design is available in the Nature Portfolio Reporting Summary linked to this article.

## Data availability

Numerical source data for graphs and plots in the main figures can be found in Supplementary Data 1. Oligonucleotide sequences used in the manuscript can be found in Supplementary Information. Additional data supporting the findings of this manuscript are available upon request. Uncropped gels and western blots from Fig. 1, Fig. 4 and Supplementary Fig. 6 are presented as Supplementary Fig. 10, Supplementary Fig. 11 and Supplementary Fig. 12.

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

## Acknowledgements

This research was funded by support from the NIHR Oxford Biomedical Research Centre (REM), the Medical Research Council [grant number MR/V029924/1] (LM), the Rhodes Trust (LEF), the Royal College of Surgeons of Edinburgh (REM), the North Harbour Club Charitable Trust (LEF), the Alain Locke Studentship, Hertford College, the Amar-Franses & Foster-Jenkins Trust (LEF, REM), The NIH [Grant R01EY034524] (JY) and the Wellcome Trust [grant number WT108369/Z/2015/Z] (REM). Elements in Figs. 1 and 4, Supplementary Figs. 6 and 7 were created in BioRender. Maclaren, P. (2024) under CC-BY licence. The authors wish to acknowledge Michelle Stewart, Gemma Codner, Sara Wells and the Mary Lyon Centre at MRC Harwell (www.har.mrc.ac.uk) for mouse services and generation of the USH2A mouse.

## Author contributions

Study conception and design: L.F., M.M. and R.M. Data acquisition: L.F., L.M., A.S., M.M., R.M., J.Y. and A.K. Data interpretation: L.F., L.M., M.M. and R.M. Editing and review of manuscript: L.F., L.M., A.S., M.M., R.M., J.Y., A.K. and L.F. had full access to all the data in the study and takes responsibility for the integrity of the data and the accuracy of the data analysis. All co-authors have reviewed and approved of the manuscript prior to submission.

## Competing interests

The authors declare no competing interests.
