## [Transparent Peer Review file · Communications Biology]

Comparison of CRISPR-Cas13b RNA base editing approaches for USH2A-associated inherited retinal degeneration

Corresponding Author: Dr Lewis Fry

Version 0:

Reviewer comments:

Reviewer #1

(Remarks to the Author)

This manuscript investigates RNA base editing approaches to correct pathogenic mutations in the USH2A gene. Loss of function mutations in USH2A cause non-syndromic retinitis pigmentosa or Usher syndrome type IIa (USHIIA), a congenital form of deaf-blindness. Specifically, the authors compare the efficiency of several CRISPR-Cas13 RNA editors in cell culture and two of these platforms (PspCas13b and Cas13bt3) in the retina in vivo. The CRISPR-Cas13 RNA editors lack nuclease activity (dCas13) and are fused with an adenosine deaminase domain (ADAR) to promote A to I conversions.

The rationale of the study is clear. Correcting disease-causative mutations in the USH2A gene/transcript is particularly desirable since gene therapy approaches based on USH2A gene delivery are limited by the large size of its coding sequence (15.6 kb). In this work, the authors focused on the c.11864G>A (p. W3955X) mutation, which creates a premature stop codon, accounts for 3-5% of USH2A cases globally, and is found in 20-84% of affected families in Europe.

The authors first tested the RNA editing efficiency in HEK293 cells using a plasmid-based dual luciferase reporter assay for A to I conversion on a 200bp mouse or human (Ush2a/USH2A) target sequence. To test the efficiency and efficacy of the proposed RNA editing approach in photoreceptors in vivo, the authors generated and characterized a novel mouse model carrying a mutation equivalent to the human c.11864G>A [mouse c.11840G>A (p.W3947X)]. The Ush2aW3947X/W3947X mice showed undetectable usherin protein in photoreceptors and cochlear hair cells and significant, measurable hearing impairment. However, no defects in retinal function or photoreceptor degeneration were observed in the Ush2aW3947X/W3947X model, limiting the evaluation of retinal phenotypic correction after RNA editing.

Overall, the study is interesting and provides information on the efficiency and specificity of several CRISPR-Cas13 RNA editors in cell culture and mouse photoreceptors in vivo. Additionally, the Ush2aW3947X/W3947X mouse model is useful for testing therapeutic approaches.

However, the rationale and conclusions of the manuscript need to be clarified and better supported by data, data description, and data analysis.

In particular:

1. The manuscript lacks important details about independent biological replicates and statistical analysis.
2. The rationale for selecting the RNA editors in the in vivo studies needs clarification (efficiency vs. specificity).
3. The observed RNA editing efficiency in vivo measured by deep sequencing needs to be better described and explained (e.g., lines 268-269 vs. lines 300-205).
4. The restoration of usherin expression in photoreceptors needs to be better reported/described. In how many eyes was usherin expression analyzed and observed?
5. The manuscript's conclusions will be strengthened if the restoration of full-length usherin after RNA editing is confirmed using a method different than immunostaining, in culture models or in vivo.
6. The data and conclusions about the potential toxicity of the Cas13 variants, gRNAs, and GFP in the retina need to be presented and described better.

Please find below the detailed point-by-point suggestions:

*Major Comments:

1. The number and sex of mice analyzed in each experiment and the number of biological replicates must be indicated in the figure and figure legends.
2. Figure 2B: the two images shown in the first top and bottom panel on the left are duplicated. Please amend.
3. Lines 229-238: The authors should describe whether auditory brainstem response (ABR) thresholds were serially analyzed over time and the onset of the auditory phenotype observed in the Ush2aW3947X/W3947X mice.
4. Figure 3F-G: the authors should indicate the number of retinas in which usherin expression was analyzed and detected by immunofluorescence using the dual vector RK-dPspCas13-E488Q and the targeting gRNA. Usherin expression can't be seen in the retinas treated with the targeting gRNA and depicted in panel F (targeting).
5. Due to the low genome editing efficiency observed in vivo having a complementary demonstration that the approach can restore full-length usherin expression would strengthen the conclusions of the study. One possibility would be to generate a surrogate cell line (e.g., HEK293) ectopically expressing the wild-type USH2A CDS and/or the mutant USH2AW3955X/W3955X CDS (e.g., using a CAG-USH2A transgene) to demonstrate the editing of the mature RNA and the restoration of full-length usherin by Western blot.
6. Lines 285-288, the authors should report the ratio on/off targeting.
7. Lines 291-294. The conclusion of the authors is not clear, please rephrase.
8. Figure 5A: A statistical analysis needs to be performed and reported.
9. Figure 5A: I suggest including the vector dose for each construct in the legend (x-axis).
10. Data depicted in supplementary figure 10 (Analysis of bystander off-target edits) should be included in Figure 4 as it shows data that are important to draw conclusions about the editing specificity.
11. Figure 5: I would suggest including the data reported in Supplementary Fig. 11 in Figure 5, to help understand the contribution of each vector to the retinal thinning. The conclusions of these analyses are not clear and should be clarified.
12. The observed RNA editing efficiency in vivo measured by deep sequencing needs to be better described and explained [e.g., 0.32-2% editing rate (lines 268-269) vs. 70-85% corrected transcript (lines 300-205)].
13. Discussion: the RNA editing efficiency and specificity observed in this study should be discussed in comparison to the previous literature on this editing technology.
14. Discussion: For the in vivo experiments, the authors investigated two approaches:
 - Co-delivering two AAV vectors, one encoding the truncated dPsp-Cas13b to fit the AAV DNA cargo capacity and the other encoding the selected gRNAs.
 - An all-in-one AAV vector approach encoding the small dCas13bt3 ortholog and the gRNA.The authors should discuss the pros and cons and future directions based on the results of their work.

*Minor Comments:

15. I would recommend including in the introduction more details about the editing technology used (e.g., the role of the mismatch in the gRNA design; current pros and cons of the approach) for the non-expert readers and providing a more detailed rationale for selecting the tested variants.
16. I suggest changing the term "in vitro" to "in culture" or "cellular models in culture" where appropriate (e.g., lines 99, 109, etc.).
17. Supplementary Fig. 1: n=2, a representation showing the standard deviation of the mean rather than the standard error of the mean would be more appropriate.
18. Lines 159-164: the described figure/panel needs to be cited.
19. Results text, Figure 1a and Legend to Figure 1a: the authors should better describe the size and sequence of the targeted USH2A/Ush2a sequence included between the Renilla and Firefly Luciferase (as reported in the methods section).
20. Supplementary Fig. 2C: the x-axis is mislabeled.
21. Line 248: The authors should comment on the selection of ADAR E488Q over the more specific, although less efficient, ADAR E488Q/T375G (Fig. 1H-K) for the in vivo study, as the specificity of RNA editing is important to prevent the generation of aberrant and/or potentially toxic proteins.
22. Lines 211-212: please specify if the antibody used for the Western blot analyses is directed to the N-terminus of the protein and to which epitope. Only an antibody binding to the usherin N-terminus would detect usherin variants truncated at the C-terminus.
23. Line 256: the authors should describe the AAV serotype and dose used in each cohort.
24. Lines 257: the authors should indicate the time point of each analysis (weeks after subretinal AAV administration).
25. Legend to Figure 2B-C. The panel description is mislabeled/missing (panel C).
26. Legend to Figure 2D. The number of mice analyzed needs to be indicated.
27. I would suggest moving the "Generation of RNA editing transgenes for AAV delivery of dPspCas13b" paragraph after the "Generation of Ush2aW3947X/W3947X mouse for RNA editing" paragraph and combining it with the description of the third AAV vector approach (dCas13bt3).
28. Figure 3 legend: abbreviations need to be spelled.
29. Figure 3D and Figure 3D legend: the author should indicate how many retinas out of total injected retinas showed dPspCas13 expression by RT-QPCR.
30. Figure 3E: please indicate the exact number of mice/group.
31. Some figures, such as figure 4 are composed of a single panel, maybe the authors may re-organize the figure content including some important data reported in the Supplementary figures (please see my suggestions in the major comments above).
32. Supplementary Fig. 6 and legend: the number of mice analyzed needs to be reported.
33. Lines 343-346: please clarify the "traditional gene replacement strategies". Dual AAV vector strategies have been proven

efficient to express full-length ABCA4 protein in the retina.
34. Line 37: Funding needs to be referred to specific authors.

Reviewer #2

(Remarks to the Author)

The discussion on W3955 in USHA2 in humans and its homologous amino acid W3947 should be revisited. If W3947 is targeted by sgRNA, it implies that the spacer used in mice would not align in humans. Therefore, it would be more optimal to develop a humanized USHA2 mouse model to enhance clinical relevance. This adjustment would significantly increase the clinical applicability of the findings.

The introduction section should be expanded to include a more comprehensive discussion on Cas13b. Currently, the text lacks detail about the enzyme. Including information on the history, origins, and mechanism of Cas13b would provide a solid foundation for its use in the study and help readers better understand its relevance and potential impact.

The sentence "Containing a C-terminally truncated PscCas13b to enable downstream AAV packaging, and ADARDD with the hyperactive E488Q mutation (Figure 1C). An A-C mismatch was encoded at the target adenosine, with 50nt gRNAs tiled across the target mutation varied by distance between the mismatched base and the gRNA scaffold" should be rewritten for clarity. Consider breaking it into simpler sentences and clearly explaining each component's role. This would make the methodology more accessible and understandable to the readers.

There is a lack of description regarding off-targeting and byproducts in figure 1. Addressing this gap is crucial to providing a complete understanding of the results. Additionally, the explanation of why off-targeting in mUsh2a outside the gRNA duplex region occurred with non-targeting guides but not with targeting guides should be elaborated on. This clarification will help readers comprehend the specificity and effectiveness of the targeting guides used.

The text currently mentions figure 3 before figure 2. Ensure that the figures are referenced in sequential order for better readability. In figure 2B, label the samples clearly, specifying what each area corresponds to and the tissue type it represents. This detailed labeling will aid in understanding the experimental setup and results.

A justification for not using a selection marker would be appreciated. If editing rates in transduced cells are believed to be up to three times higher, using a selection marker to confirm this would be beneficial. Additionally, mentioning the clinical threshold of editing necessary to prevent further degradation would provide valuable context for the study's implications.

Lastly, the study currently only targets the retina. Is it possible to also target hair follicle cells in the cochlea? Given that both the retina and the cochlea are affected, it would be interesting to see if both cell types can be edited and if there is any difference in efficiency between them. Exploring this possibility could broaden the scope and impact of the research.

Version 1:

Reviewer comments:

Reviewer #1

(Remarks to the Author)

The authors addressed most of my points and comments, and the revised manuscript is significantly improved in terms of clarity, organization, data discussion, reporting, and statistical analysis. The authors' conclusions are supported by the data and analysis, with the only caveat being the low number of eyes analyzed in some experiments, which may not be crucial at this stage of the research on RNA editing efficiency in the mouse retina ($n=2$ as reported in Fig. 3D, 3F-G).

Overall, the manuscript is significantly improved, and I suggest the following revisions:

- The authors should report RNA editing efficiency in the Abstract to clearly convey the observed results.
- The authors should include missing statistical analysis for Fig. 1 panels D-G.
- Lines 334-336 (Fig. 4A): The authors should cite the distant sites edited in the text, as these cannot be appreciated based on the red color scale presented in Fig. 4A.
- For data reported in Fig. 5A and 5B, the authors should include a statistical analysis versus the Buffer cohort.
- Statistics and Reproducibility paragraph: The authors should provide a clearer description of the statistical analysis methodology in the Methods section. Specifically, they should specify which statistical tests were used based on data distribution (e.g., normal vs. non-normal) and define the criteria for significance, including the p-value threshold (e.g., $p < 0.05$).

Minor Comments:

- The authors should revise the paper for missing words and typos. The manuscript and rebuttal contain missing references (see "Error! Reference source not found").
- Lines 126-128: The authors should specify the use of a surrogate luciferase reporter assay.
- Lines 139-140: The authors should clarify the use of either human "USH2A" or mouse "Ush2a" 200 bp target sequences in the luciferase reporter assay.
- Lines 150-151: The authors should cite Fig. 1E.

- Supplementary Fig. 1: The authors should specify the experimental system in the figure legend (e.g., HEK293).
- Lines 224-228: The authors should cite Supplementary Fig. 3 when describing the results.
- Supplementary Fig. 7: The authors should have used an RT-minus RNA retina sample as the correct control for the RT-PCR, not a no-template control (NTC). In the absence of an RT-minus control from retina samples, it is unclear if the amplified bands are derived from the episomal genomic DNA of the AAV or the AAV-derived transcript, as the sequence of the AAV genome and encoded transcript are identical.
- Lines 372-374: The authors should clarify the final part of this sentence.
- The authors should correct "Mucopolysaccharidosis type I, a lysosomal storage disease".
- Supplementary Fig. 2E: The authors should correct the question mark in the figure.

Reviewer #2

(Remarks to the Author)

All previous comments have been thoroughly addressed, and I have no further comments at this time.Thanks

Review Response

Note: line numbers in responses refer to the line numbers included in the manuscript with tracked changes

Reviewer 1 Comments

Summarised Comments

1. The manuscript lacks important details about independent biological replicates and statistical analysis.

These have been added to the manuscript, with the specific responses detailed in responses to individual comments 1, 8, 12, 9 & 30 below.

2. The rationale for selecting the RNA editors in the in vivo studies needs clarification (efficiency vs. specificity).

This has been added to the manuscript as detailed in the response to individual comment 21 below.

E488Q was chosen to demonstrate initial on-target editing as this was more efficient than E488Q/T375G in culture. T375G was not then further investigated due to the on-target editing rates found with E488Q. The following has been added to the text line 333: "All Cas13 proteins were fused to an ADAR(E488Q) deaminase domain for initial characterization as this demonstrated higher rates of on-target editing in culture."

3. The observed RNA editing efficiency in vivo measured by deep sequencing needs to be better described and explained (e.g., lines 268-269 vs. lines 300-205).

Lines 268-269 describe the overall editing efficiency detected. Lines 300-305 are describing how off-target editing affected the final transcripts. So for example, 2% of all Ush2a transcripts were corrected (lines 268-269). Of those 2%, 80% of them contained only the desired edit, with no surrounding off-target edits (lines 300-305).

Explanation altered to the following to better explain this (Line 387):
"This demonstrated that with each vector, 70-85% of the transcripts that were corrected to the wildtype sequence did not contain any additional off-target A>G edits within the region analysed."

4. The restoration of usherin expression in photoreceptors needs to be better reported/described. In how many eyes was usherin expression analyzed and observed?

As discussed in individual comment 4 below, immunohistochemistry was performed on two animals per group, in multiple areas/sections across the injected retinas. This has been added to the legend of figure 3. While the expression is low, and difficult to appreciate in the 40x image, the 63x image inset in G is provided for greater magnification.

5. The manuscript's conclusions will be strengthened if the restoration of full-length usherin after RNA editing is confirmed using a method different than immunostaining, in culture models or in vivo.

We agree with the reviewer that alternative methods of evaluation for restoration of usherin would be helpful. At this level of editing, the total amount of restored Usherin protein is low and localised to well transduced regions, and therefore is likely below the level of sensitivity of western blot. Western blot of protein from injected retina was attempted and could not detect usherin expression. Notably however, the immunohistochemistry demonstrates the protein to be precisely localised to the connecting cilium, which provides indirect evidence of restoration of the full length protein because it needs to be trafficked, folded and then inserted correctly. This requires c-terminal PDZ1-binding motif which mediates insertion into the connecting cilium membrane. This has been discussed in the manuscript, added to the discussion (line 470).

"Restoration of usherin protein was observed in areas of retina transduced with AAV, with immunohistochemistry demonstrating the protein to be precisely localised to the connecting cilium. Correct localisation provides indirect evidence of restoration of the full-length protein because it needs to be trafficked, folded and

then inserted correctly. This requires the c-terminal PDZ1-binding motif which mediates insertion into the connecting cilium membrane.⁴²”

In response to individual comment 5 below suggesting experiments to generate an USH2A-expressing culture system, while the authors agree this experiment would add further data to the paper, it is also technically challenging to create and produce plasmid and cell line that contain a full length USH2A transgene, which is very large at 15.2kb and expresses poorly. We will continue to work on developing a plasmid and cell line, however it is unlikely it will be able to be added to this manuscript.

6. The data and conclusions about the potential toxicity of the Cas13 variants, gRNAs, and GFP in the retina need to be presented and described better.

Please see responses to individual comments 8, 9, 11 below. Additional panels to figure 5 have been added to move data from the supplementary into the main text. Additional statistical analysis has been added to 5A and now 5B. Clarification in the text has been made to explain control conditions (line 423).

“Further controls were performed to isolate the cause of retinal thinning. When injected alone without a gRNA-GFP vector, the EFS-dPspCas13b-E488Q vector (1E+9 gc/eye) resulted in superior PRL thickness loss of 17 % ± 7. This was similar to the thinning produced by the gRNA-GFP vector alone at 5E+8gc/eye (16% ± 5), but much less than produced by 1E+9gc/eye (41% ± 10). This suggested the higher dose gRNA-GFP vector may cause a substantial component of the retinal thinning observed, but it was associated at some degree with all AAV preparations.”

It is challenging to accurately attribute causality of the thinning, but there is likely potential toxicity from either the AAV capsid, GFP, AAV-Cas13 or AAV-gRNA. In addition to the paragraph that already addresses this in the discussion a further comment has been added to the discussion (line 527).

“Injection of control vectors such as AAV-GFP and an AAV vector without a functional transgene would help to further understand whether the AAV capsid or GFP contributed to the retinal thinning.”

Individual Comments

1. The number and sex of mice analyzed in each experiment and the number of biological replicates must be indicated in the figure and figure legends.

These are labelled in each figure, and a comment on the approximately equal proportions of sex in each group added to the methods.

2. Figure 2B: the two images shown in the first top and bottom panel on the left are duplicated. Please amend.

Many thanks for noticing this error, amended.

3. Lines 229-238: The authors should describe whether auditory brainstem response (ABR) thresholds were serially analyzed over time and the onset of the auditory phenotype observed in the Ush2aW3947X/W3947X mice.

Thresholds were not measured serially to detect onset of the hearing phenotype. A single measurement at 9 weeks was performed. This is in the figure label, and now also added to the manuscript text.

4. Figure 3F-G: the authors should indicate the number of retinas in which usherin expression was analyzed and detected by immunofluorescence using the dual

vector RK-dPspCas13-E488Q and the targeting gRNA. Usherin expression can't be seen in the retinas treated with the targeting gRNA and depicted in panel F (targeting).

Number of animals has been added to the figure legend. While the expression is low, and difficult to appreciate in the 40x image, it is there if viewed closely, and why the 63x image inset in G is provided for greater magnification.

5. Due to the low genome editing efficiency observed in vivo having a complementary demonstration that the approach can restore full-length usherin expression would strengthen the conclusions of the study. One possibility would be to generate a surrogate cell line (e.g., HEK293) ectopically expressing the wild-type USH2A CDS and/or the mutant USH2AW3955X/W3955X CDS (e.g., using a CAG-USH2A transgene) to demonstrate the editing of the mature RNA and the restoration of full-length usherin by Western blot.

While the authors agree this experiment would add further data to the paper, it is also technically challenging to create and produce a plasmid and cell line that contains a full length USH2A transgene, which is very large at 15.2kb and expresses poorly. We will continue to work on developing a plasmid and cell line, however it is unlikely it will be able to be added to this manuscript.

6. Lines 285-288, the authors should report the ratio on/off targeting.

Added, both approximately 1:20 off/on-targets.

7. Lines 291-294. The conclusion of the authors is not clear, please rephrase.

Addressed with edits (line 375)

"This may be due to the higher expression leading to higher off-targeting rates in the RK vectors, or the low expression from EFS vectors producing off-target edits below detection sensitivity"

8. Figure 5A: A statistical analysis needs to be performed and reported.

Added

9. Figure 5A: I suggest including the vector dose for each construct in the legend (x-axis).

Added

10. Data depicted in supplementary figure 10 (Analysis of bystander off-target edits) should be included in Figure 4 as it shows data that are important to draw conclusions about the editing specificity.

Added

12. The observed RNA editing efficiency *in vivo* measured by deep sequencing needs to be better described and explained [e.g., 0.32-2% editing rate (lines 268-269) vs. 70-85% corrected transcript (lines 300-205)].

Explanation altered to the following to better explain this (Line 387):
 “This demonstrated that with each vector, 70-85% of the transcripts that were corrected to the wildtype sequence did not contain any additional off-target A>G edits within the region analysed.”

13. Discussion: the RNA editing efficiency and specificity observed in this study should be discussed in comparison to the previous literature on this editing technology.

The following paragraph has been added to the discussion to address this comment (line 477):

“While no RNA editing studies in the retina are reported, other groups have investigated RNA editing *in vivo* with various approaches. Using a miniaturised Cas13X delivered in a single AAV-PHP vector, Xiao et al. detected a $4.22\% \pm 0.68$ editing efficiency in inner ear hair cells in a model of autosomal dominant hearing loss (*Myo6*^{C442Y/C442Y}), and some recovery of hearing function in heterozygous animals.⁴³ Other site directed RNA editing systems have also been explored.

Using the MS2 bacteriophage coat protein (MCP) editing system fused to an ADAR1_{DD}, an AAV8-MCP-ADAR1_{DD}(E1008Q) construct delivered with an MS2 gRNA demonstrated 2% on-target efficiency and partial restoration of dystrophin expression in an *mdx* mouse model of muscular dystrophy.⁴⁴ Katrekar et al. employed GluR2 gRNAs which use the Q/R motif in the *GRIA2* transcript, a natural target for the binding of the dsRBD domains of full-length ADAR2.⁴⁴ They delivered ADAR2 and ADAR2(E488Q) sequences with GluR2-gRNAs via AAV8 intramuscularly to the *mdx* mouse and systemically to the sparse fur ash (*spf_{ash}*) mouse model of ornithine transcarbamylase (OTC) deficiency. This resulted in the 0.8–4.7% editing efficiency of a stop codon and splice defect in each model respectively, with resultant rescue of protein expression. In another approach, Yi et al. delivered circularised ADAR recruiting RNAs (LEAPER system) to direct endogenously expressed ADAR to edit targets in the liver of non-human primates, and the central nervous system of mice carrying the humanised *IDUA*^{W420X} mutation associated with Mucopolysaccharide Storage Disorder Type I.⁴⁵ Encouragingly, editing rates of up to 50% in the liver and 30% in the brainstem were observed. This approach is tissue dependent and requires adequate endogenous ADAR expression. A consistent feature of all studies is a significant reduction in editing efficiency *in vivo* compared to in cells in culture.”

14. Discussion: For the *in vivo* experiments, the authors investigated two approaches:

- Co-delivering two AAV vectors, one encoding the truncated dPsp-Cas13b to fit the AAV DNA cargo capacity and the other encoding the selected gRNAs.
- An all-in-one AAV vector approach encoding the small dCas13bt3 ortholog and the gRNA.

The authors should discuss the pros and cons and future directions based on the results of their work.

To address this comment this part of the discussion has been expanded with new additions underlined below (line 498):

“The overall efficiency of this approach is restricted by needing two transduction events of the same cell for both the gRNA and Cas13. In contrast to the dual vector approach, the smaller Cas13bt3 protein permitted an all-in-one AAV strategy, though editing rates were also low. Difficulties with obtaining high Cas13bt-ADAR expression and editing with an AAV2-EFS-Cas13bt1-ADAR vector have been noted by other authors.⁵ An all-in-one AAV strategy would optimise transduction efficiency and reduce overall AAV dose. Ideally future studies would employ a single vector approach, and this could be achieved using more efficient, short Cas13 effectors. Options include using alternative orthologues or through methods to improve Cas13bt efficiency and expression, such as through codon optimisation.”

15. I would recommend including in the introduction more details about the editing technology used (e.g., the role of the mismatch in the gRNA design; current pros and cons of the approach) for the non-expert readers and providing a more detailed rationale for selecting the tested variants.

The introduction has been considerably revised and expanded to address both this comment and the comments of reviewer 2.

16. I suggest changing the term “in vitro” to “in culture” or “cellular models in culture” where appropriate (e.g., lines 99, 109, etc.).

Corrected throughout the manuscript

17. Supplementary Fig. 1: n=2, a representation showing the standard deviation of the mean rather than the standard error of the mean would be more appropriate.

Adjusted, points shown

18. Lines 159-164: the described figure/panel needs to be cited.

Apologies, it is unclear what this comment refers to. Lines 159-164 refer to panel 1K which is referenced in the text.

19. Results text, Figure 1a and Legend to Figure 1a: the authors should better describe the size and sequence of the targeted USH2A/Ush2a sequence included between the Renilla and Firefly luciferase (as reported in the methods section).

Added to text (line 148): As *USH2A* expression is confined to photoreceptors and inner hair cells and not expressed in commonly cultured cell lines, a dual luciferase assay containing a 200bp target sequence of interest was developed to report A-I editing activity for screening of constructs and gRNAs in HEK293T cells (**Error! Reference source not found.A,B**).

20. Supplementary Fig. 2C: the x-axis is mislabeled.

Corrected, thank you

21. Line 248: The authors should comment on the selection of ADAR E488Q over the more specific, although less efficient, ADAR E488Q/T375G (Fig. 1H-K) for the in vivo study, as the specificity of RNA editing is important to prevent the generation of aberrant and/or potentially toxic proteins.

E488Q was chosen to demonstrate initial on-target editing as this was more efficient than E488Q/T375G in culture. T375G was not then further investigated due to the on-target editing rates found with E488Q. The following has been added to the text (Line 334): "All Cas13 proteins were fused to an ADAR(E488Q) deaminase domain for initial characterization as this demonstrated higher rates of on-target editing in culture."

22. Lines 211-212: please specify if the antibody used for the Western blot analyses is directed to the N-terminus of the protein and to which epitope. Only an antibody binding to the usherin N-terminus would detect usherin variants truncated at the C-terminus.

A C terminal antibody was used (as described in the supplementary table 3). Many available N terminal antibodies were tested with multiple protocols, but none successfully detected Ush2a expression in wildtype animals. The following is added to the manuscript (Line 254):

“ Ilyates using a C-terminal antibody demonstrated a protein band of expected size in wildtype retinae, with no band detected in *Ush2a*^{W3947X/W3947X} retinae (**Error! Reference source not found.A**)”

23. Line 256: the authors should describe the AAV serotype and dose used in each cohort.

Added to the manuscript line 335:

“All Cas13 proteins were fused to an ADAR(E488Q) deaminase domain as this demonstrated higher rates of on-target editing in culture. AAV8-Y733F vectors were delivered by subretinal injection and analyzed as shown in **Error! Reference source not found.B.**”

24. Lines 257: the authors should indicate the time point of each analysis (weeks after subretinal AAV administration).

Added to manuscript (Line 339)

“In eyes where a gRNA-GFP vector was injected, successful injection and transduction of both the targeting and non-targeting gRNA vectors was confirmed by widespread retinal GFP expression detected using *en face in vivo* imaging with blue autofluorescence (BAF) at 4 weeks post-injection (**Error! Reference source not found.C.**)”

25. Legend to Figure 2B-C. The panel description is mislabeled/missing (panel C).

Corrected

26. Legend to Figure 2D. The number of mice analyzed needs to be indicated.

Added

27. I would suggest moving the “Generation of RNA editing transgenes for AAV delivery of dPspCas13b” paragraph after the “Generation of *Ush2a*^{W3947X/W3947X} mouse for RNA editing” paragraph and combining it with the description of the third AAV vector approach (dCas13bt3).

This has been re-arranged in the manuscript as suggested

28. Figure 3 legend: abbreviations need to be spelled.

Added to figure 3 legend

29. Figure 3D and Figure 3D legend: the author should indicate how many retinas out of total injected retinas showed dPspCas13 expression by RT-QPCR.

Added to legend

30. Figure 3E: please indicate the exact number of mice/group.

Already specified in legend and displayed as individual points on each bar

31. Some figures, such as figure 4 are composed of a single panel, maybe the authors may re-organize the figure content including some important data reported in the Supplementary figures (please see my suggestions in the major comments above).

Figure 4 has been edited as per the comments above to add two more panels from the supplementary

32. Supplementary Fig. 6 and legend: the number of mice analyzed needs to be reported.

Added to legend

33. Lines 343-346: please clarify the “traditional gene replacement strategies”. Dual AAV vector strategies have been proven efficient to express full-length ABCA4 protein in the retina.

Adjusted to (line 437) “In inherited retinal disease, these account for 53% of mutations in genes such as *USH2A* and *ABCA4*, which are both common and not easily amenable to AAV-mediated gene replacement strategies due to their large size.¹⁵”. This is noting that dual-AAV strategies for ABCA4 have not progressed to clinical trial due to challenges in expression and correct expression of the split transgene.

34. Line 37: Funding needs to be referred to specific authors.

Added

Reviewer 2 Comments

1. The introduction section should be expanded to include a more comprehensive discussion on Cas13b. Currently, the text lacks detail about the enzyme. Including information on the history, size, origins, and mechanism of Cas13b would provide a solid foundation for its use in the study and help readers better understand its relevance and potential impact.

Please see extensive revision to the introduction as discussed above

2. The sentence "truncated PscCas13b to enable downstream AAV packaging, and ADARDD with the hyperactive E488Q mutation (Figure 1C). An A-C mismatch was encoded at the target adenosine, with 50nt gRNAs tiled across the target mutation varied by distance between the mismatched base and the gRNA scaffold" should be rewritten for clarity. Consider breaking it into simpler sentences and clearly explaining each component's role. This would make the methodology more accessible and understandable to the readers.

Explanations of the E488Q mutation, the C terminal PspCas13b and the A-C mismatch are now added into the introduction to make this more accessible and remove these explanations from the results. The sentences within the second paragraph of the results section (“We firstly sought to characterize the dPspCas13b-ADAR_{DD} system” onwards) have now been edited and shortened to improve clarity.

3. There is a lack of description regarding off-targeting and byproducts in figure 1. Addressing this gap is crucial to providing a complete understanding of the results. Additionally, the explanation of why off-targeting in mUsh2a outside the gRNA duplex region occurred with non-targeting guides but not with targeting guides should be elaborated on. This clarification will help readers comprehend the specificity and effectiveness of the targeting guides used.

The paragraph describing the off-targeting and byproducts for figure 1 has been edited to improve the description in the results. The interpretation of these results is limited due to these off-targets being created in a 200nt target region within a luciferase assay, and we are hesitant to over-interpret these results beyond this (Line 196).

“Finally, bystander deamination of local adenosines in the target were assessed across each construct (Figure 1K). The E488Q construct demonstrated off-target editing in both the hUSH2A (2 sites) and mUsh2a targets (8 sites). Off-targeting in the mUsh2a transcript outside the gRNA duplex region also occurred in corresponding positions with the non-targeting guide, which did not occur in the hUSH2A target. This suggests there may be characteristics of the target mUsh2a sequence or structure that increases the likelihood of off-target deamination even in the absence of a double stranded gRNA-target complex. Comparatively, the E488Q/T375G deaminase mutant was more specific, editing at only the target adenosine in both targets, with no off-target editing in with non-targeting guides. No editing was detected in samples transfected with targeting guides only in the absence of Cas13-ADAR.”

4. The text currently mentions figure 3 before figure 2. Ensure that the figures are referenced in sequential order for better readability. In figure 2B, label the samples clearly, specifying what each area corresponds to and the tissue type it represents. This detailed labeling will aid in understanding the experimental setup and results.

The order of the figures in the text has now been adjusted with the movement of the “Generation of RNA editing transgenes for AAV delivery of dPspCas13b” section further down the manuscript. Additional labels within Figure 2 have been added to demonstrate the tissue type, and layers within each tissue for clarity.

5. A justification for not using a selection marker would be appreciated. If editing rates in transduced cells are believed to be up to three times higher, using a selection marker to confirm this would be beneficial. Additionally, mentioning the clinical threshold of editing necessary to prevent further degradation would provide valuable context for the study’s implications.

Cell sorting of retinal cells using e.g. FACS or MACS techniques is technically challenging resulting in a significant loss of tissue material and low yields of RNA

from already small tissue samples. We agree that given the low rates of editing detected, it would have been beneficial in this case to have sorted and selected for GFP+ cells to gain a true idea of the editing rate, however this would require significantly larger animal numbers to achieve sufficient tissue. The following comment has been added to the discussion (Line 465): “As this rate is derived from processing of whole retina, editing rates in transduced cells may be up to 3-times higher, as only approximately 30% of the retina is transduced via a single subretinal injection. Further studies to isolate GFP positive populations of cells using cell sorting could confirm this.”

The clinical threshold for editing rates required remains unknown, but it is postulated to be low. The discussion on this has been expanded to discuss a similar retinal cilial protein CEP290 (line 557):

“USH2A is an excellent therapeutic target for a base editing approach. It is a very large and stable structural protein expressed in the photoreceptor cilium. Restoration of a similar large cilium protein CEP290 via RNA editing with antisense oligonucleotides demonstrates restored ciliogenesis in pre-clinical models,⁶⁰ and durable visual improvements in patients following intravitreal administration.⁶¹ This supports the notion that whilst the corrected RNA half-life might be low, large structural proteins may last for some time in the cilium, and may have a far more durable response than the RNA itself. Only a small restoration of expression may be required to enable gradual accumulated restoration of the protein. Further longitudinal studies will be required to determine how long the expressed vector continues to edit the target, and the half-life of usherin following repair.”

6. Lastly, the study currently only targets the retina. Is it possible to also target hair follicle cells in the cochlea? Given that both the retina and the cochlea are affected, it would be interesting to see if both cell types can be edited and if there is any difference in efficiency between them. Exploring this possibility could broaden the scope and impact of the research.

Yes, it is also possible to target IHC cells in the cochlea, with other DNA base editing approaches demonstrating this in other diseases. However, this is less clinically relevant for Usher syndrome type 2, as patients are born congenitally deaf with non-progressive hearing loss, unlike the ocular aspect of the disease which is adolescent-adult onset progressive retinal degeneration. It is likely that Usherin is required for the development and correct localisation of proteins within the IHC ankle-link in the pre-natal/early post-natal cochlea. In mice Usherin expression is not detectable beyond the early post-natal (PND15) period in WT animals, and it is unlikely to have a role in mature hair cells. As it is unlikely that patients will be identified early enough to treat the developing cochlea, a therapeutic approach to this was not pursued.

Review Response, Second Revision

18/11/24

Reviewer 1

- The authors should report RNA editing efficiency in the Abstract to clearly convey the observed results.

The following text has been added to the abstract:

“Mean RNA editing rates in photoreceptors across different constructs ranged from 0.32% to 2.04%, with greater efficiency in those injected with PspCas13b compared to Cas13bt3 constructs. In mice injected with PspCas13b constructs, usherin protein was successfully restored and correctly localized to the connecting cilium following RNA editing.”

- The authors should include missing statistical analysis for Fig. 1 panels D-G.

This has been added to the graph to show significant editing relative to the NT guide and the figure legend updated. We note for the editors that we do not believe this improves the quality of the analysis, as the purpose of this data is to select highly performing guides relative to the non-targeting guides which is clearly demonstrated. Adding statistics with asterisks here unnecessarily adds complexity to the figure.

- Lines 334-336 (Fig. 4A): The authors should cite the distant sites edited in the text, as these cannot be appreciated based on the red color scale presented in Fig. 4A.

As detailed in the figure legend, the significantly edited distant sites are already highlighted clearly in black outline, and do not need further citation in the text.

- For data reported in Fig. 5A and 5B, the authors should include a statistical analysis versus the Buffer cohort.

This figure has been updated along with the caption and the associated text to show comparison of significant differences between the superior retina and buffer groups.

The following has been added at line 360-364:

“Comparing the superior retinal thickness of the injected groups compared to buffer, only the EFS-dPspCas13b-E488Q (targeting) and RK-dPspCas13b (targeting and non-targeting) showed significant thinning ($p < 0.05$).”

- Statistics and Reproducibility paragraph: The authors should provide a clearer description of the statistical analysis methodology in the Methods section. Specifically, they should specify which statistical tests were used based on data distribution (e.g., normal vs. non-normal) and define the criteria for significance, including the p-value threshold (e.g., $p < 0.05$).

The statistics and reproducibility section has been updated.

Statistics and group sizes are described for each figure. Data were assessed for normality and analysed using standard statistical tests for normal and non-normal data as described for each figure using GraphPad Prism (version 10) software. Comparisons between groups were made by assessing effect sizes, and significance determined with a p value of < 0.05 where appropriate. Each replicate represents an individual measurement from a separate sample.

- The authors should revise the paper for missing words and typos. The manuscript and rebuttal contain missing references (see "Error! Reference source not found").

This has been checked thank you. There appears to be an error introduced in the PDF conversion process when it was uploaded. This has been fixed.

- Lines 126-128: The authors should specify the use of a surrogate luciferase reporter assay.

Updated

“We firstly compared the dPspCas13b-ADAR¹² and the dCas13bt-ADAR¹⁶ systems in a surrogate luciferase assay in cells in culture for the repair of mutations in both *USH2A* and *Ush2a*.”

- Lines 139-140: The authors should clarify the use of either human “USH2A” or mouse “Ush2a” 200 bp target sequences in the luciferase reporter assay.

Added.

“As *USH2A* expression is confined to photoreceptors and inner hair cells and not expressed in commonly cultured cell lines, a dual luciferase assay containing a 200bp target sequence of interest from *USH2A* and *Ush2a* was developed to report A-I editing activity for screening of constructs and gRNAs in HEK293T cells (**Error! Reference source not found.A,B**).”

- Lines 150-151: The authors should cite Fig. 1E.

Added.

“The limited number of 30nt guides screened showed lower editing efficiency relative to the 50nt guides (**Error! Reference source not found.E**).”

- Supplementary Fig. 1: The authors should specify the experimental system in the figure legend (e.g., HEK293).

Added

The dual-luciferase assay conducted in HEK293T cells (as in figure 1) was performed to compare gRNAs that had the hairpin-loop direct repeat (DR+) to facilitate interaction with Cas13, and with gRNAs with no direct repeat (DR-). For both PspCas13b targeting either the mUsh2a target or the hUSH2a target (A, B) or with Cas13bt3 targeting mUsh2a (C), editing activity was abolished by removal of the direct repeat from the gRNA. Data shown as mean ± SD, n = 2.

- Lines 224-228: The authors should cite Supplementary Fig. 3 when describing the results.

This was already cited in line 226, but an additional citation is now placed at line 229.

“Although no usherin protein was detected, *Ush2a* mRNA expression levels from retinal lysates did not demonstrate a difference between wildtype and *Ush2a*^{W3947X/W3947X} mice (**Error! Reference source not found.**). However, when using probes specific to either the wildtype or c.11840 G>A mutation, no wildtype *Ush2a* was detected in *Ush2a*^{W3947X/W3947X} and similarly no W3947X transcripts were detected in the wildtype, consistent with sequencing of RT-PCR products (**Error! Reference source not found.**).”

- Supplementary Fig. 7: The authors should have used an RT-minus RNA retina sample as the correct control for the RT-PCR, not a no-template control (NTC). In the absence of an RT-minus control from retina samples, it is unclear if the

amplified bands are derived from the episomal genomic DNA of the AAV or the AAV-derived transcript, as the sequence of the AAV genome and encoded transcript are identical.

Thanks for this comment. An RT-ve control was not run on this gel, but prior to this experiment a control was run using qPCR in a separate prior experiment and demonstrated similar RQ values to no template controls.

- Lines 372-374: The authors should clarify the final part of this sentence.

Amended to:

This analysis suggests that the high dose (1E+9) gRNA-GFP component of the dual vector injections were the most toxic component of the dual vector system.

- The authors should correct "Mucopolysaccharidosis type I, a lysosomal storage disease".

Amended to

In another approach, Yi et al. delivered circularised ADAR recruiting RNAs (LEAPER system) to direct endogenously expressed ADAR to edit targets in the liver of non-human primates, and the central nervous system of mice carrying the humanised *IDUA*^{W420X} mutation associated with Mucopolysaccharidosis type I, a lysosomal storage disease.⁴⁵

- Supplementary Fig. 2E: The authors should correct the question mark in the figure.

The symbol error creating a question mark in supplementary figure 6E has been corrected. Thank you.